# Evidence of vertical coupling: Meteorological storm Fabienne on 23 September 2018 and its related effects observed up to the ionosphere.

Petra Koucká Knížová[1], Kateřina Podolská[1], Kateřina Potužníková[2], Daniel Kouba[1], Zbyšek Mošna[1], Josef Boška[1] and Michal Kozubek[1]

[1]Department of Ionosphere and Aeronomy, Institute of Atmospheric Physics, Boční II/1401, 14100 Prague 4
[2] Department of Meteorology, Institute of Atmospheric Physics, Boční II/1401, 14100 Prague 4

*Correspondence to*: Petra Koucká Knížová (pkn@ufa.cas.cz)

**Abstract:** Severe meteorological storm system on the frontal border of cyclone Fabienne passing above Central Europe was observed on 23–24 September 2018. Large meteorological systems are considered to be important sources of the wave-like variability visible/detectable through the atmosphere and even up to ionosphere heights. Significant departures from regular courses of atmospheric and ionospheric parameters were detected in all analyzed data sets through atmospheric heights. Above Europe, stratospheric temperature and wind significantly changed in coincidence with fast frontal transition (100–110 km h$^{-1}$). Zonal wind at 1 and 0.1 hPa changes from usual westward before storm to eastward after storm. With this changes are connected changes in temperature where at 1 hPa analyzed area is colder and at 0.1 hPa warmer. Within ionospheric parameters, we have detected significant wave-like activity occurring shortly after the cold front crossed the observational point. During the storm event, both by Digisonde DPS-4D and Continuous Doppler Sounding equipment, we have observed strong horizontal plasma flow shears and time-limited increase plasma flow in both North and West components of ionospheric drift. Vertical component of plasma flow during the storm event is smaller with respect to corresponding values on preceding days.

Analyzed event of exceptionally fast cold front of the cyclone Fabienne fell into the recovery phase of minor-moderate geomagnetic storm observed as a negative ionospheric storm in European Mid-latitudes. Hence, ionospheric observations consist of both disturbances induced by moderate geomagnetic storm and effects originated in convective activity in troposphere. Nevertheless, taking into account significant change in global circulation pattern in the stratosphere, we conclude that most of the observed wave-like oscillations in the ionosphere during night 23–24 September can be straight attributed to the propagation of atmospheric waves launched on the frontal border (cold front) of the cyclone Fabienne. Frontal system acted as an effective source of atmospheric waves propagating upward up to the ionosphere.

# 1     Introduction - Variability of the ionosphere

The ionosphere is highly variable system that is influenced by solar and geomagnetic activity from above and lower-laying atmospheric phenomena from below. Ionospheric variability is observed on a wide-scale range from minutes, or even shorter, up to scales of solar cycle and secular variations of solar energy input. With no doubt, the most dominant driver of ionospheric variability is solar activity. Whole atmosphere and ionosphere react according to the level of solar energy input. The episodes of limited strongly enhanced dissipation of solar energy (solar flares, coronal mass ejections etc.) can affect only regions localized in high latitudes or can cover all the geosphere. During such event, the magnetosphere is affected first (see for instance Hargreaves, 1992). A large portion of solar energy is dissipated in the upper atmosphere and then in the ionosphere, thermosphere (Davies, 1990; Solomon and Qian, 2005). The perturbations can be detected at the ground level, for instance by magnetometers. Disturbances associated with such enhanced solar energy inputs are in general called geomagnetic storm (Gonzales et al., 1994, Buonsanto, 1999) or geospheric storms (Prölss, 2004). Different types of solar agents that are mainly responsible for geomagnetic disturbances have been analyzed with respect to their geoeffectiveness by Kakad et al. (2019), Georgieva et al. (2006), Fenrich and Luhmann (1998), Leamon et al., (2002), Prölss (2004) and many others.

Besides the solar and geomagnetic forcing the energy inputs from lower-laying atmospheric heights must be taken into account in the energy budget of the ionosphere. The lower laying atmosphere and its impact on the ionosphere have been largely studied during last exceptionally low solar cycle by mean of growing number of satellite measurements. Paper Anthes (2011) and more recent paper Liu at al. (2017) demonstrated effectivity of Radio Occultation (RO) sounding methods on board of satellites for systematic sounding of the atmosphere with respect to weather, climate and space weather.

The ionosphere is weakly ionized plasma where both neutrals and ions play an important role. Ionization degree around maximum of electron concentration is less than $10^{-2}$ and significantly smaller below the maximum in the F layer, except limited events of Sporadic E layer occurrence (Whitehead, 1961,1990; Mathews, 1998; Haldoupis, 2012). The impact of the collision processes on the ionospheric dynamics cannot be neglected especially in the lower ionosphere. During day time, due to incoming solar radiation, ionosphere is formed at height of mesosphere and thermosphere. Ionosphere is typically stratified into D, E and F layer, where the maximum electron concentration is usually located. The F layer is usually a region with maximum electron concentration. It can be split into two sub layers denoted F1 and F2 layers. In case of splitting into F1 and F2 layers, the maximum of electron concentration is located in F2 layer. During night time electron concentration decreases at all heights due to recombination processes and lack of ionizing radiation. It leads to practical disappearance of all ion pairs below F layer that remains present due to slow recombination processes at its height (Davies, 1990; Rishbeth, 1998; Prölss, 2004 among many others). As a measure of the ability of the Earth´s atmosphere to absorb incoming solar radiation we can consider the maximum electron concentration NmF2 in the highest ionospheric level F or F2 if present. During solar cycle, we can observe clear link between incoming solar radiation and ionospheric ionization. With the

increasing solar activity we observe higher ionization. However, the relationship is not linear and is subject of large investigation. The link between ionospheric variability and both solar and geomagnetic indices were analyzed for instance by Clilverd et al., 2003; Cnossen et al., 2014; Forbes et al., 2000; Roux et al., 2012; Koucká Knížová et al., 2018, Perrone et al., 2017. Understanding of the relation between solar activity and corresponding ionospheric and/or atmospheric behavior is crucial for instance in the estimation of the trends and potential human impact on the atmosphere and ionosphere (Roininen et al., 2015; Laštovička et al., 2012; Laštovička, 2012; Georgieva et al., 2012).

Ionosphere clearly reflects solar activity on all studied time-scales. Diurnal courses of the maximum concentration in the ionosphere clearly show the dominant solar influence, increase/decrease of the electron concentration with respect to solar zenith angle. During stable solar and geomagnetic situation, however, significant difference in the courses of ionospheric parameters is well seen on consequent days. Vertically propagating gravity waves are subject of large scientific interest since 1960s. A fundamental interpretation of atmospheric variability in terms of atmospheric gravity waves was provided by Hines (1960) and later by Hines (1963, 1965, and 1968 among others). The effects of gravity waves on in the ionosphere up to F2 region through photochemical and dynamical processes were discussed by Hooke (1970b). Garcia and Solomon (1985) reported GW importance on the chemical composition of the middle atmosphere. There, it has been already shown that the resulting effects of gravity waves depend not only on the wave properties but on the actual ionospheric situation and/or direction of propagation with respect to incoming solar radiation (Hooke, 1970a; 1971). It has been pointed out by Holton (1983) that gravity wave drag and diffusion are fundamental for the wind and temperature balance in the middle atmosphere. Fritts and Nastrom (1992) suggested that convective activity in the troposphere is as important source of gravity waves as topographic forcing. Model study of gravity wave generation and its observable signatures above deep convection is provided by Alexander et al. (1994). Tropospheric convective systems are often connected with strong lightning. Possibility of thunderstorm influence on ionosphere has been already suggested by Bhar and Syam (1937). In general, two principal mechanisms are proposed. First mechanism presumes gravity waves generated by thunderstorm to propagate up to ionospheric heights. Second mechanism involves generation of electrical discharges in the E region above the storm. Applying superposed epoch analyses, Davis and Johnson (2005) reported statistically significant intensification and decent in altitude of midlatitude sporadic E layer directly above thunderstorm. Different observational result showing decrease of critical frequency of sporadic E has been reported by Barta et al. (2017). Mechanism involved in the coupling between thunderstorm lightning and ionosphere is very complicated and not well understood yet. The limitations of generally accepted mechanisms are discussed in detail in the paper Haldoupis (2018).

Later detail model studies of gravity wave propagation through the Earth's atmosphere simulations provided by Vadas and Fritts (2005), Vadas (2007), Vadas and Nicolls (2012) proved that gravity waves originating in the tropospheric convection can reach thermospheric heights and significantly affect wind and temperature profiles. Atmospheric waves propagate from lower laying atmosphere up to the thermosphere as primary waves or dissipate. The deposited momentum excites secondary waves (see for instance Vadas and Liu (2009) or Vadas et al. (2018)). Atmospheric vertical coupling via propagation of internal atmospheric waves was largely investigated by Yigit and Medvedev (2017). Their model study revealed significant impact of gravity waves on mean circulation and cooling the thermosphere down by up 12-18%. Further, they demonstrated influence of gravity waves on tides both direct and indirect.

The effects of GW on the atmosphere up to thermospheric heights are highly variable due to complexity of the system.

Except model studies there are observational evidences of consequent ionospheric disturbances attributed to dynamical processes in the lower atmosphere. Chernigovskaya et al. (2018) provides an evidence of F2-layer ionospheric response to dynamic processes during the winter circumpolar vortex evolution in the strato-mesosphere. McDonald et al., (2018) reported an enhancement in total electron content in ionosphere, which coincides with the commencement of a stratospheric warming event. Goncharenko et al. (2010) observed persistent variations in the low-latitude ionosphere that occur several days after a sudden warming event in the high-latitude winter stratosphere. Enhancements of wave-like activity within ionospheric F layer with relation to meteorological events were reported by Chernigovskaya et al. (2015). Propagation of concentric gravity waves from source region in the troposphere related to tropospheric convective storm up to the ionosphere was reported by Azeem et al. (2015). Paper presents almost simultaneous observations of a gravity wave event in the stratosphere, mesosphere, and ionosphere. Suddenly increasing wave-like oscillations within ionospheric parameters after passing tropospheric cold front across observational point was reported by Boška and Šauli (2001) and Šauli and Boška (2001). On the longer term-term scale, the extremely high correlation between ionospheric measurements of the up to the 'break point' at 10 degrees in longitude and/or Earth´s distance 1000 km is attributed to the mesoscale systems as proposed by Koucká Knížová et al. (2015). Infrasound waves excited by severe tropospheric storms (e.g. typhoons and strong storms) are discussed and analyzed. Chum et al. (2018) detected infrasound in the ionosphere from earthquakes and typhoons, by mean of Multipoint Continuous Doppler Sounding equipment. Authors give examples of observation by an international network of continuous Doppler sounders. The waves were observed at the height range from about 200 to 300 km by continuous Doppler sounder located in Taiwan (Chum et al., 2018). The infrasound was observed during several hours for strong storms events.

GPS satellite measurements are promising tools for monitoring ionospheric changes connected with severe weather systems. Recently the analyses of scintillation $S_4$ index in relation to four tropical cyclones (Yasi in 2011, Marcia in 2015, Debbie in 2017 and Marcus in 2018) were presented by Ke et al. (2019). They found intensification of scintillation effects mostly above the tropical cyclone path and attributed them to the electric field perturbation and consequent plasma bubble generation. Within COSMIC GPS data, Yang and Liu (2016) has found significant peak in radio occultation scintillation events during the passage of tropical cyclone Tembin (2012) during quiet geomagnetic or solar aktivity and attributed the observed effect to the gravity waves generated in the lower atmosphere by the cyclone. Afraimovich et al. (2013) published large review of GPS/GLONASS studies of the ionospheric response to natural and antropogenic processes and phenomena. Paper focuses on wide range of ionospheric forcing and corresponding ionospheric variability detected in principle within Total Electron Content (TEC) and F2 layer critical frequency foF2. In relation to tropical cyclones (Katrina, Rita and Wilma) occurring in 2005 they reported increase of wave-like activity in gravity-wave period range mainly in the range 20 to 60 minutes and intensification of TEC variations along the satellite path close cyclone. The zones of disturbances were found to form during hurricane stage of the cyclone (Afraimovich et al., 2013).

Review of lower atmosphere forcing was provided by Lastovička (2006). Detail insight into internal wave coupling processes in Earth's atmosphere comprising teoretical, model and experimental recent results can be found in Yigit and Medvedev (2015) and later in Yigit et al.

(2016). The importance of involvement of lower atmosphere into ionospheric variability study in order to accurately capture smaller-scale features of the upper atmosphere response even to the geomagnetic storms, is demonstrated by Pedatella and Liu (2018). The evidence of lower atmosphere forcing is clearly demonstrated on the day-to-day ionospheric variability (known as an ionospheric anomaly) during low and stable solar and geomagnetic activity during consequent days. Ionospheric parameters (e.g. electron concentration or height of ionospheric layers) on such scales are influenced by combination of meteorologic activity and solar/geomagnetic forcing. During geomagnetically quiet days the tropospheric forcing is more emphasized and relatively more important and is ruling the ionospheric dynamics, far more than the solar and geomagnetic energy inputs.

Model study (Pedatella, 2018) demonstrated variability of the response of the atmosphere and ionosphere system to one particular storm when the internal variability characterized by the ensemble standard deviation is introduced. The study shows that implementation of arbitrary internal atmospheric variability leads to the geomagnetic storm occurring under a different, though climatically similar, atmospheric state for each ensemble member. The study has found that variability leads to uncertainty typically 20%−40% with localized regions exceeding 100%. It clearly shows that large-scale features of the storm are reproduced well and while small-scale characteristics of the response are dependent on lower atmosphere variability. Hence neglecting of the lower atmosphere may lead to significant complication in the geomagnetic storm interpretation.

## 2    Data

For the description of cyclone Fabienne in the troposphere we use **meteorological** ground-based data (https://www.ventusky.com/, www.wetterkontor.de, http://wetter3.de, http://www.ufa.cas.cz/institute-structure/department-of-meteorology/present-weather-sporilov.html) and Aeolus satellite measurements described in the following chapter 2.1. Behavior of the stratosphere is interpreted using **stratospheric** wind and temperature reanalysis MERRA 2 datasets (https://disc.gsfc.nasa.gov/daac-bin/FTPSubset2.pl) described in chapter 2.2. The ionosphere observation (details are provided in chapter 2.3) comes from two ground based vertical **ionospheric** sounding using the Digisonde DPS 4D (http://giro.uml.edu/ and http://digisonda.ufa.cas.cz/ ) and oblique reflection using the multipoint Continuous Doppler Sounding (CDS) http://www.ufa.cas.cz/files/OHA/M_Doppler_system.pdf. Besides that we use satellite TEC measurement (http://gnss.be/Atmospheric_Maps/ionospheric_maps.php) for station Pruhonice. For geomagnetic situation description we use geomagnetic indices from Potsdam Data Center https://www.gfz-potsdam.de/en/kp-index/. The data used for interpretation of Fabienne event and related disturbances in stratospheric and ionospheric heights cover time interval 20–27 September 2018.

### 2.1    Meteorological data

In order to describe severe storm Fabienne we use ground-based meteorological monitoring combined with satellite observation. For determination of the synoptic condition in

the troposphere, surface and upper synoptic maps were used (available at https://www.wetterkontor.de/ and http://wetter3.de). We also used meteorological ground-based radar observations taken from the https://www.ventusky.com/. In addition hourly averages meteorological data performed by automatic weather station located at the Institute of Atmospheric Physics IAP (50.04°N, 14.48°E) were used for determining the time of frontal passage (http://www.ufa.cas.cz/institute-structure/department-of-meteorology/present-weather-sporilov.html). Data are available for last 30 days, and then they are stored in the institute archive.

The Earth Explorer Atmospheric Dynamics Mission Aeolus yields data from global observations of wind profiles from space using the active Doppler Wind Lidar (DWL) method (Gompf, 2000). The DWL measurement is the unique method that has the potential to provide the required data on a global scale, from direct observation of wind. The DWL measures 100 wind profiles per hour using both Rayleigh and Mie scattering method (Durand et al., 2004). The global wind profiles (along a single line-of-sight) are measured up to an altitude of 30 km to an accuracy of 1 m s$^{-1}$ in the planetary boundary layer (up to an altitude of 2 km). The Aeolus mission was launched on 22 August 2018 and scientific measurement started on 12 September 2018.

## 2.2 Stratospheric Data

The MERRA2 (Modern-Era Retrospective analysis for Research and Applications, version 2 from https://disc.gsfc.nasa.gov/daac-bin/FTPSubset2.pl) with resolution 0.5° in latitude and 2/3° in longitude was used. The MERRA-2 is a global atmospheric reanalysis produced by the NASA Global Modelling and Assimilation Office (GMAO), details can be found in Gelaro et al. (2017). The MERRA2 is available up to 0.1 hPa from 1980 till present but we show only 1 and 0.1 hPa for period from 20 September 2018 to 27 September 2018 which is relevant for our studies. This reanalysis provides reliable time series in regular grid network. Temperature and zonal wind 6-hourly data (00, 06, 12 and 18 UT) in the stratosphere and lower mesosphere (from 30–80 km) were used. The MERRA 2 reanalysis has many advantages as reliable time series without gaps, regular grid network or high vertical resolution. Of course there are some disadvantages. Because the reanalysis includes many observations datasets from satellite, radiosondes or ground measurements they have to be assimilated into one dataset. That is why we can get biased dataset especially at higher altitudes. However, usage of MERRA 2 for our analysis is sufficient as the reanalysis provides us with clear description of the stratosphere situation during the studied interval.

## 2.3 Ionospheric Data

State of the ionosphere has been monitored globally on a regular base since setting of the network of ionosondes in frame of the International Geophysical Year in 1957–1958. Some of the ionospheric station are still operating and represent observatories with longest time series of ionospheric data available for research.

Vertical sounding of the ionosphere is based on the reflection of electromagnetic wave from ionospheric plasma. Sounding pulse is reflected from plasma unit when the sounding frequency is equal to its plasma frequency (see for instance Davies (1990)). Using typical sounding frequency range 1 MHz–20 MHz it is possible to monitor ionosphere from the E layer up to maximum electron concentration in the F region. With increasing frequency of the sounding wave, the pulse penetrates higher to the ionosphere. When the frequency of the sounding pulse exceeds plasma frequency of maximum, the pulse propagates through the ionosphere without reflection and no echo is registered in the receiver. Maximum frequency of the reflected wave from the particular layer is called critical frequency and is simply related to maximum plasma concentration of the layer. For the purpose of the analyses we use maximum of electron concentration NmF2 located in the F or F2 layer, and the corresponding plasma frequency called critical frequency and denoted foF2. Time series of foF2 are the longest data sets available for systematic study of ionospheric variability.

In the Observatory Pruhonice (49.9°N, 14.6°E) located close to Prague, Digisonde DPS 4D is used for regular ionospheric monitoring. Digisonde DPS 4D provides ionograms, directograms and skymaps for further evaluations and interpretations. Digisonde operates in the multi-beam sounding mode using six digitally synthesized off-vertical reception beams in addition to the vertical beam. For each frequency and height on a multi-beam ionogram, the raw data from the four receive antennas are collected and processed to form seven beams, separately for the O-mode and X-mode echoes (Reinisch, 1996; Reinisch et al., 2005). Detail description can be found also on web page http://umlcar.uml.edu/digisonde.html.

All the data were manually checked and evaluated. Detail processing of the drift measurement and how the skymaps are controlled, is described by Kouba at al. (2008) and Kouba and Koucká Knížová (2012). High-rate sounding campaign partly overlaps our selected time span. The aim of the high-rate sounding measurement was to monitor short-term variability of the Es-layer. Hence, our data consists of data with 2-minutes (till 24 September at 6:30 UT) and 15-minutes repetition time. Ionospheric drift data are not yet widely used for description of ionospheric variability. Kouba and Koucká Knížová (2016) have provided first systematic study of regular course of vertical drift component in Mid-latitudes. The study was conducted during year 2006, i.e. during time interval described by low solar and geomagnetic activity. It shows diurnal and seasonal variability of the vertical plasma drift component and quantifies its characteristic values.

Ionogram represents height-frequency characteristics of the ionosphere above the station. It displays virtual reflection height vs. sounding frequency. Using only vertical echo on the ionogram one can receive height profile of frequency or electron density (Davies, 1990 and many others). According to the receiving antenna field, multi-beam ionograms can be recorded. Digisonde can register off-vertical reflections in addition to the vertical one. The off-vertical signals are further processed to show characteristics of the oblique reflection caused by ionospheric irregularities. Directogram (see for more detail description http://ulcar.uml.edu/directograms.html) provides information about direction of the echoes received from irregularity. The central column between the panels corresponds to the vertical

reflection at zero zenith angle. Shades of blue in the directogram correspond to general direction of plasma-drift from west to east, and shades of red are used to represent drift in the opposite direction, i.e. from east to west.

In addition to Digisonde DPS-4D, ionosphere is regularly monitored by a multi-point continuous Doppler (CDS) sounding portable system based on the measurements of the Doppler shift experienced by waves reflected from the ionosphere. The measurements are simultaneously performed on 3 to 5 frequencies with 4 Hz separation around the center frequency of 3594.5 kHz. Multipoint measurement makes it possible to investigate propagation of infrasonic waves or ionospheric oscillations caused by fluctuations of geomagnetic field etc. (http://www.ufa.cas.cz/files/OHA/M_Doppler_system.pdf). Observation of wave propagation in the ionosphere is performed on the basis of multi-point and multi-frequency continuous Doppler sounding (CDS) in the Czech Republic. We used two multi-point CDS systems operating at frequencies of 3.59 and 4.65 MHz. Kouba and Chum (2018), demonstrated efficiency of Digisonde-based drift measurement together with Continuous Doppler Sounding on fixed frequency for study of dynamics of the ionosphere. Chum et al. (2018) detected infrasound waves generated by seven typhoons that passed over Taiwan or in its surroundings in period 2014–2016. The spectral characteristics of the ionospheric infrasound from convective storms are sensibly similar as for the cotyphoon infrasound. The highest spectral densities were observed during about 2–5 minutes (3.3–8.3 mHz).

The ground-based ionospheric sounding was complemented by Total Electron Content (TEC) above station Pruhonice derived from satellite measurement (http://gnss.be/Atmospheric_Maps/ionospheric_maps.php). While the foF2 parameter describes local maxima of electron concentration, and thus the variation of foF2 can be attributed mostly to the ionization-recombination processes in the F2 region the TEC satellites measurement is a parameter representing integral of electron concentration from bottom to upper part of the ionosphere.

## 3    Meteorological description of storm Fabienne

**Synoptic evolution:** On 20 September 2018 a short-wave travelled eastward from the British Isles towards Sweden and Poland and supports strengthening of a cyclone Elena over the North Sea. Anticyclonal warm late summer condition with weak southwest flow occurred in Central Europe. A strong frontal zone from the north Atlantic over the Central Europe separated cool Atlantic air from the hot continent. The unusually hot and dry weather in the Central Europe culminated September 21 afternoon and accelerate the movement of cold front around noon. Along the front there was forming squall line with thunderstorms in the evening. Recorded strong wind gusts were caused by both convective activity and a significant pressure gradient within the cyclone.

During the following days, the low descent centre moved towards the northeast and formed a deep cyclone above Scandinavia. Because of the strong zonal flow along the lower edge of this cyclone another front system coupled with the Fabienne cyclone quickly moved to Central Europe. On 23 September the cyclone Fabienne deepened and passed through Central Europe to the east. Within the warm sector ahead of a cold front of the Fabienne humid low-level air is advected northeastward from the subtropical North Atlantic. This resulted in

evolving intense convection and formation squall line with thunderstorms along the very fast moving cold front. Synoptic time-evolution is demonstrated in the Figure 1.

**Surface data:** In Figure 2, there are average hourly data measured at Institute of Atmospheric Physics (IAP) meteorological station clearly show the cold front that passed over this station at 18 UT on 21 September, when ground level pressure reached a local minimum. Before the front, maximum air temperatures at 2m reached tropical values above 30 °C, behind the front the maximum daily temperature did not reach more than 20 °C. The average hour wind speed intensified before the front and during the rainfall associated with storm activity. After passing the front, the surface wind changed from southwest to northeast.

Around 15 UT on September 23, the warm front brought light rain associated with stratiform clouds. The temperature at 12 UT on September 23 was lower than in the afternoon when the area was temporarily in the warm sector of cyclone Fabienne. Lifetime of the warm sector was very short. The time-series of surface variables at IAP station show the warm front which is connected with slight direction windshift, but rapid rise in temperature until 18 UT on September 23. At this time the passage of the cold front connected with Fabienne occurred and brought a thunderstorm activity with heavy rain and wind shock. The surface pressure minimum was close to 1000 hPa, while the temperature reached local maximum and the wind direction changed from from west to north. The hourly mean wind speed rose rapidly up to midnight. On September 24 was cold, the maximum temperature reached only 12.8 °C. The strong cold northwest wind remained across the measurement point, the maximum averaged hourly values reached 7 m.s$^{-1}$ in the afternoon. Pressure continued to rise until midnight following day. Centre of the massive anticyclone Shorse (see Figure 1), which moved from the British Island over the Czech Republic in just 36 hours brought the pressure to a value of 1040 hPa.

Both the Europe surface pressure charts in Figure 1 and the ground level time series for IAP observatory in Figure 2 displays unusually fast passage of synoptic pressure patterns over the Central Europe. Cyclone Fabienne moved at around 25 m.s$^{-1}$. It exceeds speeds of extratropical transition. Jones et al. (2003) described extratropical transition of tropical cyclones which can accelerate from a forward speed of 5 m.s$^{-1}$ in the Tropics to more than 20 m.s$^{-1}$ in the Mid-latitudes. Sanders (1986) noted mean surface cyclone speed of moving of about 18 m.s$^{-1}$ for cyclone originated in the west-central North Atlantic and deepened explosively. A manual in Czech language based on both empirical observations and classical synoptic meteorology states that the average cyclones speed over Europe is around 8 m.s$^{-1}$ to 11 m.s$^{-1}$ (Kopáček, Bednář, 2009)

The synoptic-scale windstorm connected with the cyclone Fabienne is an unusual event for the time of occurrence (23 September 2018) and for the storm moving velocity and intensity (Kašpar et al, 2017). The month of September in the Middle Europe is typically characterized by significant condition under high pressure, i.e. relatively weak wind sunny days. Figure 3 exhibited the fast moving of cyclone Fabienne in a strong zonal flow from Atlantic region across the Central Europe. Within the warm sector of the cyclone unstable wet air has been advected from subtropical Atlantic region to northest-ward perpendicular to the direction of cyclone moving. Follow-up strengthened baroclinity of the atmosphere at lower levels was the main cause of quick cyclone deepening (visible on surface pressure field - white lines) and generated storms at the head of the cold front.

The 500 hPa map shows the main flow regime of the troposphere. (The atmosphere at an altitude about 5.5 km is no longer under the influence of surface friction. In synoptic meteorology 500 hPa map is used to determine the speed and direction of synoptic patterns.) In the morning on 23 September the density of isohyps depicted at 4-decameters interval indicated a large pressure gradient between the warm southern and cold northern parts of Europe. At 12 UT the cyclone center is still located above Germany at 18 UT it is above the territory of the Czech Republic. On the 850 hPa pseudo-equivalent potential maps there are clearly visible narrow transformation zones with a strong gradient of pseudo-equivalent potential temperature. These "warm boundaries" are separated various homogenous air masses with different temperatures and locate the position of the fronts on the surface pressure field Kašpar (2003) (http://www.met.wur.nl/education/atmospract/unit9/thetaw%20and%20fronts.pdf). From radar images presented on Figure 4, the speed of the squall line can be estimated at 110 km/h. Impacts of the strong wind gusts associated with this squall line passage have been well documented on the European Severe Weather Database (https://www.eswd.eu).

**Satellite Aeolus observation provides** global wind profiles (along a single line-of-sight) up to an altitude of 30 km with an accuracy of 1 m s$^{-1}$ in the planetary boundary layer (up to an altitude of 2 km). The Aeolus mission was launched on 22 August 2018 and scientific measurement started on 12 September 2018. All data outputs (including ALADIN instrument) have already been verified and their reliability verified for the examined period. The Earth Explorer Atmospheric Dynamics Mission Aeolus yields data from global observations of wind profiles from space using the active Doppler Wind Lidar (DWL) method (ESA, 1989; Durand et al., 2004). The DWL measurement is the unique method that provides data on a global scale from direct observation of wind. The Aeolus Doppler Wind Lidar measures 100 wind profiles per hour using both Rayleigh and Mie scattering method (for more information see https://earth.esa.int/web/guest/missions/esa-operational-eo-missions/aeolus).

The graphs in Figure 5 display the wind profiles measured by Aeolus ESA satellite using ALADIN instrument working at 355 nm during the time period from 22–24 September 2018 (orbit numbers 481 to 520), geographical coordinates ranges: 12°–19° E, 48°–51° N; geomagnetic coordinates: 97° L, 48° F, -17° Y. The vertical axes represent altitude of height bins, while the horizontal axes represent time of observation. Data catalogue is provided by ESA EO, from http://aeolus-ds.eo.esa.int/socat/L1B_L2_Products. The accuracy is limited by the design of the instrument. In all the comparisons we consider this aspect. Single Doppler Wind Lidar is able to measure both Mie scattering from particles and aerosols, and Rayleigh scattering from the upper atmosphere molecules. This study uses the Rayleigh scattering measurement with random error (1σ) is 1 m.sec-1 at altitudes less than 2 km, 2 m.sec$^{-1}$ between 2 and 16 km. Systematic error (1σ) is in this case smaller than 0.7 m.sec$^{-1}$ (Durand et al., 2004).

In the two upper panels of Figure 5a and 5b we may see situation before the storm on 22 September. At heights above 10 km there is area where the satellite registers opposite direction of the wind compare to surrounding regions (marked by blue color). Figure 5c represents situation of early morning of 23 September before the cyclone Fabienne has entered the area of measurement site. Calming of the windflow caused by temperature daily cycle is clearly visible. Figure 5d shows the post storm effect on 24 September. The area of opposite wind direction detected by satellite Rayleight scattering is lifted up to heights of 15 km. The

measurements at the time of Fabienne storm passage above the measurement site is not available due to satellite trajectory, however from the satellite records before and after the cyclone passage indicate extremely high speed changes within troposphere and lower stratosphere.

## 4     Stratospheric dynamics 20–27 September 2018

Stratosphere and its dynamics are very sensitive for wave activity in higher or lower layers (troposphere or mesosphere). That is why a strong storm Fabienne as a source of many different kind of waves should bring disturbance into regular dynamics. With changes in dynamics are connected changes in temperature and vice versa. Stratospheric wind and temperature for Europe region are presented for time span 20 September at 00 UT to September 27 at 18 UT (each day is represented by one row on the Figures). This period covers whole week (3 days before and 4 days after the Fabienne storm).

Figure 6a shows zonal wind at 1 hPa for Europe region from 20 September at 00 UT to 27 September at 18 UT. On the sequence there is well seen weak eastward wind in middle Europe and westward wind in south Europe which is typical situation for this period. Shortly before storm Fabienne (23 Sep 00 and 6 UT) easterly wind became stronger and replace westerly wind in the south (because of incoming waves from troposphere) and remain easterly for the following several days in whole studied area. At 0.1 hPa we can see changes from westerly to easterly wind shortly after Fabienne (24.9. 00 and 06 UT). We do not register any significant changes before because wave from the troposphere need some time to reach 0.1 hPa. Strong easterly wind remains in whole Europe again for several days after storm. The stratosphere needs some time for changing/restoration dynamics to normal situation because of wave disturbances which remains in inversion condition (temperature increase with altitude) much longer than in other layers. That is why we can observe strong eastward wind not only during the storm but for several days after storm in whole Europe as well.

The changes in the zonal wind, which mainly control stratospheric dynamics (meridional wind is much weaker but very important for Dobson-Brewer circulation), are usually connected with changes in temperature because strong zonal wind effectively block air mixing form different latitudes especially in higher latitudes. The temperature is observed directly so we can expect better approximation in reanalysis than for zonal wind which is derived parameter. The temperature fields at 1 hPa are presented on Figure 7a. We can find much colder air in middle and northern Europe shortly before and mainly after storm Fabienne (23 September 06 and 12 UT) because usual strong barrier (different winds between high and lower latitudes) are destroyed after Fabienne and much colder air from polar vortex can reach lower latitudes in our case middle and south Europe. Colder air in lower latitudes for several days after storm Fabienne could impact not only chemistry in higher stratosphere in autumn but mesosphere condition as well (i.e. the propagation of waves could be slower or faster than usual or they can be absorbed in the stratosphere). This change is well connected with change of zonal wind.

At 0.1 hPa (on the Figure 6b) the change of the zonal wind is even stronger than at 1 hPa (compare Figure 6a and b) because the wind in the mesosphere is usually stronger than in the stratosphere.  As was pointed in wind results, several hours after storm (24 September 06 and 12 UT) zonal wind changes from westward or very weak eastward to strong eastward and remains without changes for several days. Meridional wind in mesosphere is stronger than in the stratosphere but still zonal wind plays major role in the dynamics changes. Temperature changes at this level, that corresponds to lower mesosphere, are opposite from pressure level 1 hPa. There is warmer air in higher latitudes so stronger eastward wind brings this warmer air to the lower latitudes (because it is not blocked by the westward wind) and affects the whole area of the middle Europe. The warmer air stays in analyzed area because we need to wait until the zonal wind reverses to his usual state (westward instead of eastward). We have to notice that 0.1 hPa is above stratosphere and the dynamics here could be affected by different processes (i.e. solar radiation, chemistry etc.) than at 1 hPa (almost stratopause).

Especially 0.1 hPa are on the top of the MERRA2 reanalysis so we should be very careful with interpretation of the results because the information from this level could be affected by border condition or problem with wave activity dissipation. But we need mainly qualitative description rather than quantitative description for our study so information from MERRA2 are sufficient.

## 5    Geomagnetic Situation – Preceding moderate storm

September 2018 was a period of rather low geomagnetic activity. On 4 September the geomagnetic activity increased for about 20 hours. Maximum registered Kp index was Kp = 6. The following period was characterized by low geomagnetic activity with Kp up to 3 till September 18 when the Kp index fall down to 0 and remain very low till September 21. On September 21 the geomagnetic activity increased again at 22:30 UT when Kp = 4+. Increased geomagnetic activity lasted for about 20 hours with maximum Kp = 5- on 23 September at 03 UT. The activity can be classified as a minor to moderate geomagnetic storm. On 23 September, the geomagnetic activity falls down again to values around Kp = 2 to 2+. In general, there was rather low geomagnetic activity with only short slightly enhanced events. However, the geomagnetic effects are responsible for part of the observed ionospheric variability and cannot be completely neglected. Geomagnetic indices were downloaded from Potsdam Data Center https://www.gfz-potsdam.de/en/kp-index/. Detail plot of geomagnetic Kp index for time span 21-26 September is provided as a part of Figure 10 (bottom panel).

## 6    Ionospheric dynamics and wave activity

An example of multi-beam ionograms measured by DPS-4D is shown in the Figure 8. There is a sequence of ionograms recorded during four consequent days around 23 UT. The receiving antenna system of the digisonde is able to identify the direction of the electromagnetic wave arrival. The information about the reflected wave arrival is included in the raw ionograms. Further, from sequence of raw ionograms the general plasma motion is constructed and presented as the directogram. Each color corresponds to particular antenna beam, hence the

direction of the arrival of oblique echo from large scale irregularities. Colors of echo indicate particular direction of arrival. It is clearly shown that echo changes significantly. Ionogram at 23 UT is selected to show changing dynamics of the ionospheric plasma. During night time, ionograms with clear echo are typically recorded. Antenna system registers practically only vertically reflected signal, as it is shown on panel (a) measured on 22 September and panel (d) recorded on 25 September. In comparison, qualitatively different pattern is detected on panels (b) and (c). The echo on panel (b), recorded on 23 September shortly after passage of the cold front with heavy storm activity, is called spread F situation. As it is indicated by color scheme on the right side of each ionogram, antenna system records echo from practically all sounding beams. Both vertical and off-vertical echoes are spread in height and frequency. It means that the ionosphere is full of irregularities and iso-contours of electron concentration are significantly undulated. On the panel (c) measured on 24 September, there is well recorded vertical echo and slightly higher oblique structure reflected from North-North-East direction. Such kind of echo is known as spur echo and may appear when ionospheric iso-contours are significantly tilted. In general, vertical echo on panel (c) corresponds to situation on panels (a) and (d). Spread F situation on panel (b) indicates significant wave-like activity within ionospheric plasma in the F-region.

Sequence of directogram measurements recorded by DPS-4D is shown in Figure 9. There are two distinct episodes of increased activity. On the directograms measured first two days 21–22 September, wave activity is rather low. Situation changes significantly on 23 September at 17 UT when strong echo is recorded till 24 September at 4 UT. Signal detected by the receiver varies significantly during night. An interesting fact is that DPS-4D instrument detects strong quickly changing plasma shears and reversal plasma motion with respect to zero zenith-angle. There is no prevailing or characteristic plasma flow for the event. During day-time, there is very low activity visible on the directogram till evening hours. Strong echo is recorded again from 24 September at 17 UT till 25 September at 4 UT. However, the echo is not as strong as the preceding night with smaller shears. Prevailing or dominant plasma motion during local night 24–25 September is in North-North-East direction.

On the plot of the diurnal course TEC in Figure 10 (upper panel), decrease can be observed on 23–24 September compared to previous day September 22$^{nd}$. Similarly, in Figure 10 (middle panel), decrease in critical frequency foF2 was observed during 23–24 September. While TEC and foF2 show significant decrease in reaction to minor geomagnetic disturbance, there is no clear change in course and shape of critical frequency of E layer foE Figure 10 (middle panel) except of very short wave-like variability on 23 September before Fabienne storm passage above the observational site. On 23 September, maximum of foE reaches same values as on preceding and following days. Most of the variability is observable within time series of TEC and foF2 and both parameters agree well through the studied interval. Their matching can be explained by dominant contribution of F2 layer's electron content contribution to the TEC and much less contribution of E layer's variability during studied days, even during the Fabienne event. The effect of electron concentration decrease in the ionosphere can be attributed to the geomagnetic disturbance observed as a negative storm effect (Prölss, 2004) related to the decrease of the atomic oxygen leading to decrease of production of oxygen ions

and the increase of molecular nitrogen density leading to the increase of loss rate of ion species. Both processes lead to electron-ion concentration decrease. Values of critical frequencies foF2 and TEC return to typical values of the season comparable with those preceding the observed geomagnetic storm event on 25 September (two days after the storm passage over Pruhonice station). Geomagnetic disturbance started on 21 September at 21 UT. Frequency foF2 during night falls much faster than it is typical. Then during night 22-23 September, critical frequency foF2 falls even faster, oscillates and remains below 3.5 MHz till almost noon when rapidly increases. During night 23–24 September, after sunset critical frequency foF2 decreases faster compared to nights 21–22 September and 25–26 September, when typical course of foF2 is registered.

The observed variability of the parameters TEC, foF2 and foE on Figure 10 is caused jointly by the minor geomagnetic disturbance and atmospheric waves associated with Fabienne storm. It is practically impossible to distinguish what part of the variability belongs to the particular forcing. Ionospheric vertical sounding has, unfortunately, limitations and provides integral information about resulting behavior of the atmosphere. However, in addition to the time of flight of the electromagnetic wave the DPS 4D equipment recorded additional parameters of the reflected wave from ionosphere. Variability of critical frequency foF2 must be interpreted together with complete ionogram record. As it is demonstrated in the Figure 8, there is well seen change of the ionogram pattern through experiment. Ionograms recorded on 22 September (type on panel a) are usually recorded when the reflection plane is practically flat while ionograms recorded on 23 September (type on panel b) are measured when reflection planes are significantly undulated. Such situations occur in association with atmospheric wave activity. Hence, this additional information can be used to slightly untangle effects of the geomagnetic disturbance and convective activity. Taking into account the course of foF2 and TEC together with change of ionogram reflection patterns, we suppose, that dominant effect of the geomagnetic disturbance is pronounced as decrease of foF2 and TEC, while short term wave-like variability around mean course associated with spread echo occurrence on ionograms can be attributed to the convective activity in the lower atmosphere.

All ionograms were manually scaled and further used for determination of vertical profile of electron concentration (or frequency) with the use of NHPC inversion technique that is part of the digisonde software. Details and downloads are available on web page: http://umlcar.uml.edu/digisonde.html. In agreement with course of foF2 and TEC, analyses of entire electron density profiles reveal the same decrease on 23–24 September as a consequence of the preceding geomagnetic moderate storm. In order to illustrate well the electron concentration variability related to cold front effect we focus on profilograms for three consequent days 22–24 September.

Due to geomagnetic disturbance electron concentration and corresponding plasma frequency decrease which leads to problematic representation of the situation for entire time-span 21–26 September. Figure 11 shows variations of reflection height of the sounding signal recorded on 22–24 September for selected range 2–6 MHz with 0.1 MHz step. Oscillations in heights clearly show strong wave-like activity within all ionospheric heights. Comparing

reflection heights at fixed frequencies for two consequent nights, there are shorter period oscillations visible during night on 22–23 September compared to night-time on 23–24 September. Oscillations detected during both nights are coherent through all levels.

Similar effects of oscillation as observed on Figure 11 are seen on the detail plot of profilograms Figure 12 composed from true-height profiles during all analyzed days. Deviation from regular course is well seen on profile thickness that is significantly smaller on 23 September till 24 September about 6 UT, when the thickness of profiles increases again.

Further we have analyzed critical frequency foF2 using continuous wavelet transform to obtain power content on particular periods. Wavelet Power Spectrum (WPS) for 21–26 September is presented on the Figure 13. Oscillations on shorter periods on 22 September compared to 23 September are well seen in Figure 13. In the plot of WPS, there are high power domains of short-period oscillation in the range 5–30 minutes during day-time on 21 September and 22 September. Less energy is detected during day-time on 23 September. Missing spectral content on periods below 30 minutes on 24–26 September is caused by change of sounding rate on 24 September at 10 UT. As it has been explained in data section, high sampling rate campaign was switched till morning on 24 September for study of Sporadic-E phenomenon.

Following three panels in Figure 14 show the ionospheric drift evolution 20 September–27 September. In the plots of North (panel b) and East (panel c) components, during several days preceding both geomagnetic and meteorological storm there are only rare situations where the ionospheric plasma motion was detected in a horizontal plane. Episode of longer duration of plasma flow in the horizontal plane is detected after sunset on 23 September when the storm Fabienne hit observation point. Characteristic value of plasma flow velocity is $v_{North} \sim 40$ m.s$^{-1}$ and $v_{East} \sim -30$ m.s$^{-1}$. Comparing North component of the plasma flow during night of the Fabienne storm (24 September, at 1 UT–4 UT) and corresponding time following days, it is important to point out that the flow is in opposite direction and practically same velocity magnitude.

Vertical components in Figure 14 (a) show typical diurnal course with two minima, one located close to sunrise and one close to sunset (same as reported by Kouba and Koucká Knížová, 2016). Values of vertical drift before Fabienne storm event reached regularly larger values with respect to days after the event. For instance, magnitudes of sunrise negative velocity peaks are detected around $\sim -50 - \sim -30$ m.s$^{-1}$, while after the event sunrise peaks are not exceeding $\sim -20$ m.s$^{-1}$. The abrupt change is seen on 23 September at 19 UT, soon after the cold front passing above the observation point. Characteristic values before storm are exceeding $v_v \sim 20$ m.s$^{-1}$, while after the storm they hardly reach $v_v \sim 20$ m.s$^{-1}$ and rather stay close $v_v \sim 10$ m.s$^{-1}$. Change in plasma flow is well pronounced in all three drift velocity components shortly after the frontal passage above observational point at ground level.

In the following Figure 15, we show Continuous Doppler Sounding (CDS) measurement on three consequent days 22–24 September on frequency 3.59 MHz (a) and 4.65 MHz (b). Beginning the storm Fabienne (or passage above observational point) is visible in the data as a short-duration increase of noise across the CDS spectra on both frequencies, Qualitative change of the echo is evident for the first sight. Data were obtained from Continuous Doppler Sounding

spectrogram archive IAP CAS, Prague, http://datacenter.ufa.cas.cz/. Spectrograms of the recorded infrasound during event Fabienne until 4 UT correspond to the reference time for this event.

The spectral content changed with time and was different during the strong storm event compared to preceding and following day. During afternoon hours on 22 September, CDS registers clear sharp echo with wave-like fluctuations. On 23 September on both frequencies we have observed sudden increase of noise at 18 UT that could indicate arrival of acoustic wave packet from the frontal border. After that, stronger and blurred echo compared to 22 September is registered on both frequencies. Wave-like fluctuations are not detected within the signal on 3.59 MHz and 4.65 MHz. On both frequencies (better pronounced on 4.65 MHz on Figure 15 b), there are apparent coincidental drops in frequency at 18 UT. Blurred strong echo was observed until around 4 UT on 24 September. In the afternoon hours on 24 September, recorded CDS echo remains slightly blurred but it is significantly weaker. The occurrence of stronger echo on CDS sounding on 3.59 MHz in the interval 18 UT (23 September) till 4 UT (22 September) corresponds to the increased wave activity on directograms and detection of plasma flow on both North and East plasma drift components. The trace of 4.65 MHz is limited due to diurnal course of foF2. Hence the changes of in the CDS signal can be discussed only till 20 UT. Signal detected on 23 September is significantly stronger with respect to preceding and following days, especially in the part that corresponds to the frequency drop at 18 UT.

According to our experience in Chum et al. (2018) we can conclude that we observe disturbances related to waves propagating from lower-laying atmosphere. It was shown that the cotyphoon infrasound waves were recorded in the spectral range from ~3.5 to 20 mHz with maximum of spectral density around 5 mHz (dominant periods between 3 and 4 minutes). The spectra revealed fine structures that were likely caused by modal resonances.

**7      Conclusion**

We have analyzed atmospheric and ionospheric effects induced by fast transit of cold front with strong storm of the cyclone Fabienne. Cold front passed above Europe within 24 hours with high speed reaching values 30 m.s$^{-1}$ (approx. 108 km.h$^{-1}$) on 23 September 2018. The synoptic-scale windstorm connected with the cyclone Fabienne was untypical in its time of occurrence, and velocity of storm moving activity highly exceeded standard values for the season. The temperature drop on frontal border of 10 °C is also rather large. The major damages were caused by the storms on frontal border mainly on the territory of Germany, where the wind gusts reached extreme values 45 m.s$^{-1}$ (162 km.h$^{-1}$). In the Czech Republic, the strongest wind gusts of about 35 m.s$^{-1}$ (126 km.h$^{-1}$) were recorded mountains. Significant strong wind was observed in lowlands as well. For instance, in the meteo-station Karlov, located in Prague, wind gust reached values 27 m.s$^{-1}$ (97 km.h$^{-1}$).

We have detected significant change in the dynamical pattern in stratosphere followed immediately after storm both in wind and temperature. General circulation pattern above Europe at 0.1 hPa before the storm Fabienne event can be classified/characterized as part of the

stratosphere in normal condition in September. Based on that, we attribute the overall change of the stratospheric circulation/dynamics to the strong wave-field that was launched upward from the fast moving mesoscale system.

At the time of Fabienne event, ionosphere was slightly influenced by minor to moderate geomagnetic storm that occurred one day before. According to the evolution of Kp index and ionospheric plasma parameters (TEC and foF2) ionosphere was already in the recovery phase of the geomagnetic storm. Nevertheless, the observed disturbances are induced both by geomagnetic storm and convective activity in the lower laying atmosphere. Regarding results of model study (Pedatella, 2018) we attribute general decrease in foF2 and TEC to the geomagnetic forcing (longer-term, negative storm scenario) and significant increase in wave-like activity (short-term, wave-like activity) to the convective system forcing.

We have found significant departures from typical values of ionospheric parameters shortly after transition of the cold front across the observation point. We have detected sudden strong increase of wave-like activity on the directograms and CDS records. Detected strong echo on directograms shows strong and rapid changes in the horizontal plasma motion. During the observation, there was no prevailing plasma motion direction. It rather accounts for turbulent flow within F-layer. In the strong echo in directograms attributed to the storm, there is no characteristic prevailing motion, but sudden changes in direction are observed through the event. Time-limited increase of plasma drift in North and East direction has been detected together with decrease of velocity of the vertical plasma flow. Wave-like oscillations are present within ionospheric plasma all the time. In the WPS spectra of critical frequency we have detected change of the spectral content during day of the Fabienne event compared to preceding day. We have noticed decrease of F layer thickness during day of the Fabienne event. Irregular stratification of the ionosphere is confirmed by spread-echo recorded by Digisonde during afternoon and night on 23 September till morning 24 September. CDS data show significant change in spectral content, shape and power of the registered signal corresponding to modulation by waves propagating from convective system. As it has been pointed out, for instance in Yigit and Medvedev (2015, 2017), the gravity waves originating in the lower atmosphere are able to significantly alter processes up to the thermospheric heights and their impact is highly variable according to actual state of the background atmosphere.

On the above summarized results we conclude that mesoscale system Fabienne is effective source of atmospheric disturbances that can reach ionospheric heights and significantly alter atmospheric and ionospheric conditions. Convective system Fabienne affected Earth's atmosphere on a continental scale and up to F-layer heights. Even during periods of geomagnetic disturbance, minor to moderate geomagnetic storm, the contribution of the lower atmosphere to the ionospheric dynamics cannot be neglected. Our experimental result is in agreement with theoretical study of Pedatella (2018) that internal atmosphere variability should be taken into account even during geomagnetic disturbances.

*Code availability*

http://umlcar.uml.edu/digisonde.html

*Data availability*

Data used for the paper can be downloaded via following sites:
https://www.ventusky.com/
https://www.wetterkontor.de/
http://wetter3.de,
http://www.ufa.cas.cz/institute-structure/department-of-meteorology/present-weather-
sporilov.html
http://aeolus-ds.eo.esa.int/socat/L1B_L2_Products
https://disc.gsfc.nasa.gov/daac-bin/FTPSubset2.pl
http://giro.uml.edu/
http://digisonda.ufa.cas.cz/
http://www.ufa.cas.cz/files/OHA/M_Doppler_system.pdf
http://datacenter.ufa.cas.cz/
http://gnss.be/Atmospheric_Maps/ionospheric_maps.php
https://www.gfz-potsdam.de/en/kp-index/

*Team list*
Petra Koucká Knížová, Kateřina Podolská, Kateřina Potužníková, Daniel Kouba, Zbyšek
Mošna, Josef Boška and Michal Kozubek

*Author contribution*
Petra Koucká Knížová (PKK) – DPS 4D ionogram data scaling, analyses and interpretation,
first draft of the paper
Kateřina Podolská (KAPO) – AEOLUS data analyses and interpretation, CDS data analyses
and interpretation.
Kateřina Potužníková (KACA) – meteorology data preparation and interpretation
Daniel Kouba (DK) – DPS 4D settings, Drift data analyses
Zbyšek Mošna (ZM) - TEC data analyses and interpretation, figure preparation
Josef Boška (JB) - DPS 4D data interpretation
Michal Kozubek (MK) – stratospheric data analyzes and interpretation

*Competing interests.*

Petra Koucká Knížová is one of the Special Issue Editors Annales Geophysicae.

*Acknowledgements:*

Authors would like to thank colleagues from the Institute of Atmospheric Physics CAS Marek
Kašpar for synoptic interpretation of Fabienne cyclone peculiarities, and further Jaroslav Chum,
Jiří Baše and František Hruška for maintenance of Doppler systems. Authors would like to
acknowledge Global Ionosphere Radio Observatory (GIRO) and its mirror-site hosted in the
Institute of Atmospheric Physics CAS.
Work of PKK, DK, ZM, JB was supported by the H2020 COMPET-2017 TechTIDE Project.
Work of KAPO was supported by project 18-01969S and work of MK by project 18-01625S
of Czech Science Foundation.

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

1<br/>2<br/>3<br/>4<br/>5<br/>6<br/>7<br/>8

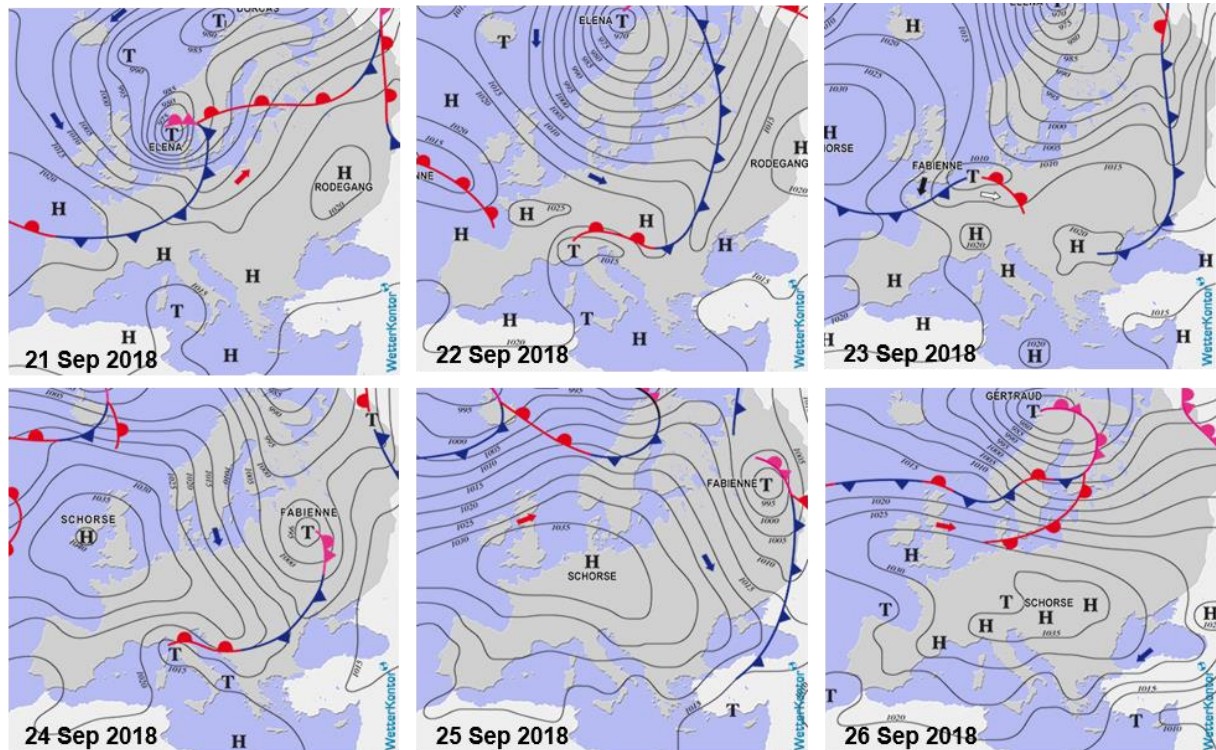

9<br/>10<br/>11

**Fig. 1.** Surface pressure maps provided by Wetterkontor, from [www.wetterkontor.de](www.wetterkontor.de). Surface pressure is plotted with solid lines with 5 hPa step. Atmospheric fronts (red curved lines with red semi-circles that point in the direction of warm front, blue curved line with blue triangles that point in the direction of cold front and purple line with alternating triangles and semi-circles pointing in the direction in the occluded front is moving), the location of the centres of high (H) and low (T) pressure systems are also presented.

18<br/>19<br/>20<br/>21<br/>22<br/>23<br/>24<br/>25<br/>26<br/>27<br/>28<br/>29<br/>30

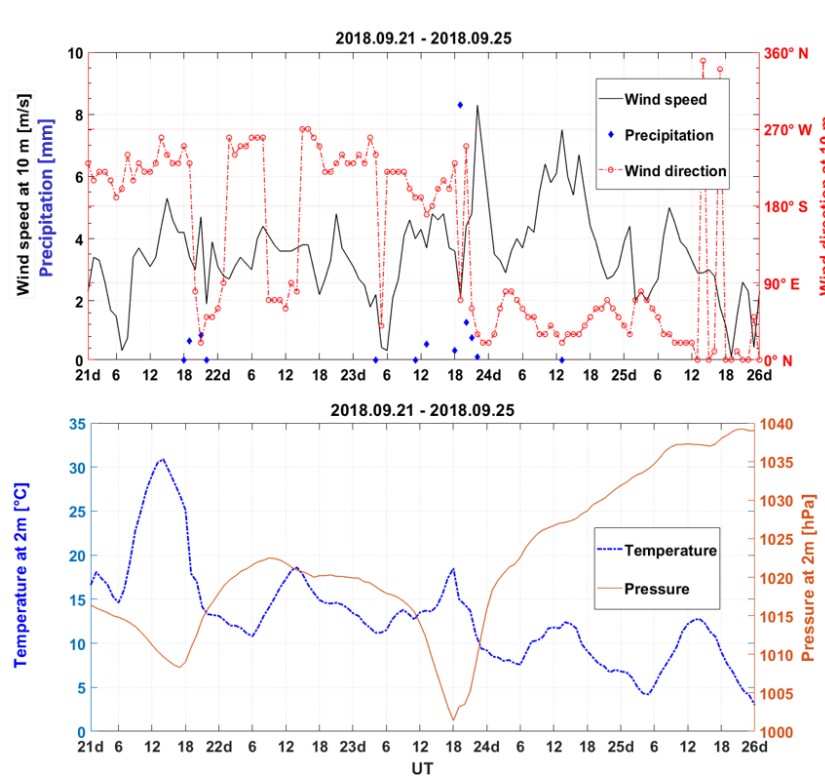

**Fig.2.** Hourly averages surface observations at IAP meteorological station. Atmospheric pressure, air temperature and precipitation amount are measured at 2 m height above the surface, wind speed and direction at 10 m height above the surface.

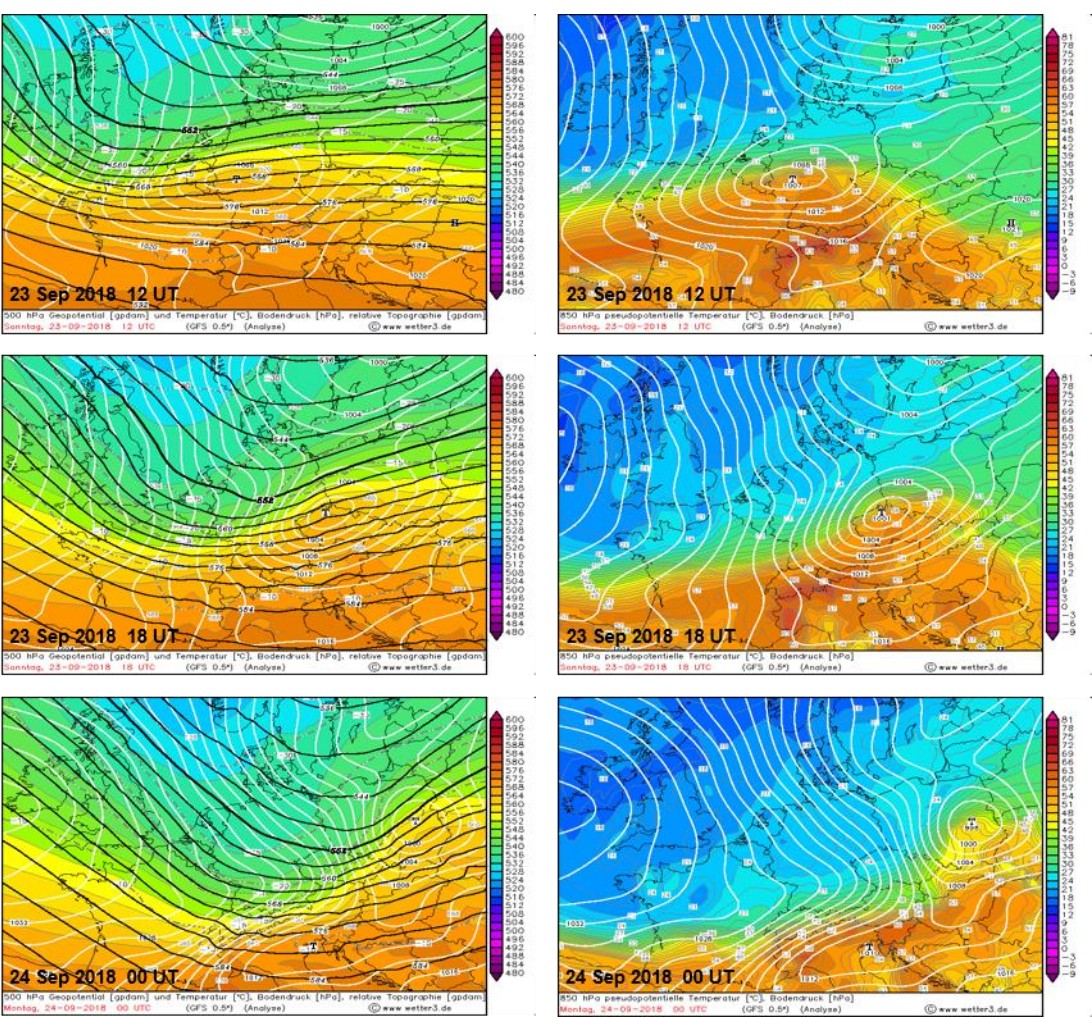

**Fig.3. Left panels:** Distribution of the geopotential field (black lines) and temperature (gray
dashed lines) at level of 500 hPa, of the surface pressure field (white lines), and of relative
topography 500–1000 hPa (color field). The 500 hPa is given in units of 10 geopotential
decameter (gpdam), the temperature in °C, the surface pressure in hPa, and the 500/1000 hPa
thickness in gpdam.  Isohypses and thickness are 4 gpdam apart, isobars 2 hPa and isotherms 5
11 °C. The thickness or difference in heights between the 1000 hPa (surface) and 500 hPa levels
varies on temperature and moisture (is a function of average virtual temperature), thus the color
field regions depicted the average temperature of the troposphere. (Orange/red values indicate
warm tropical air, blue/indigo indicate artic air.) **Right panels**: Analysis of pseudo-equivalent
potential temperature in 850 hPa pressure level in °C (color field and isotherms - gray lines, 3
16 °C apart) and surface pressure field (isobars - white lines, 2 hPa apart). The pseudo-equivalent
potential temperature is conservative in association with the moisture thus it allows to compare
temperature of air masses in lower troposphere regardless of their humidity.

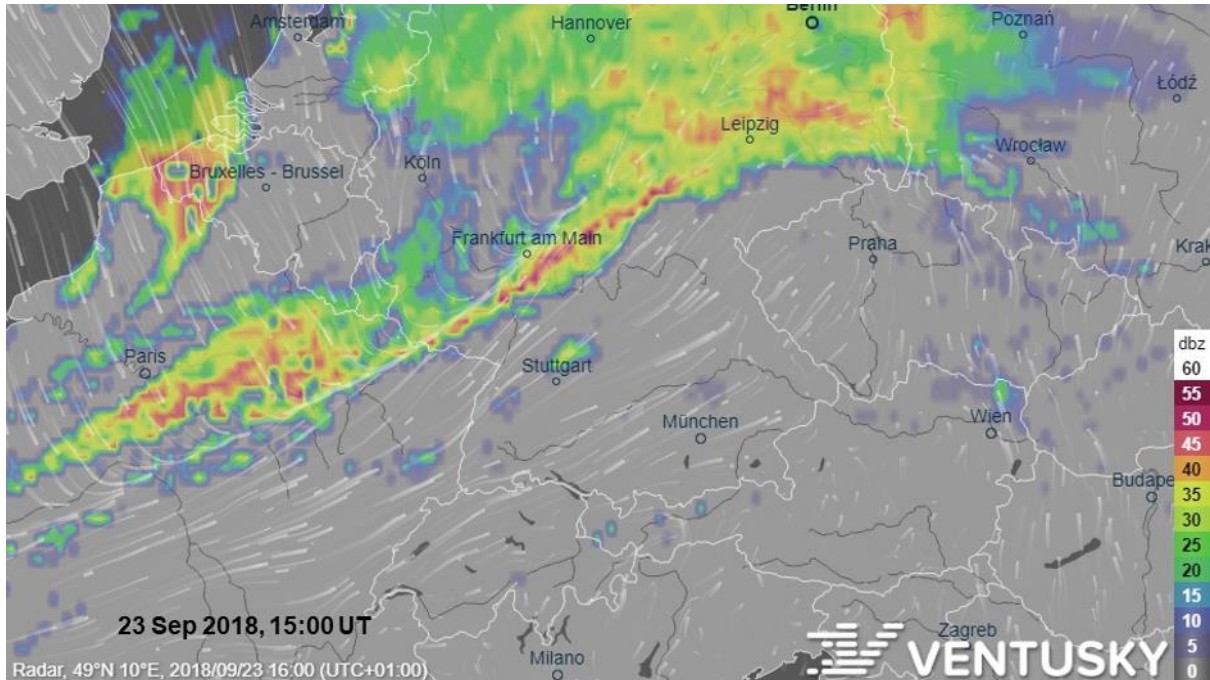

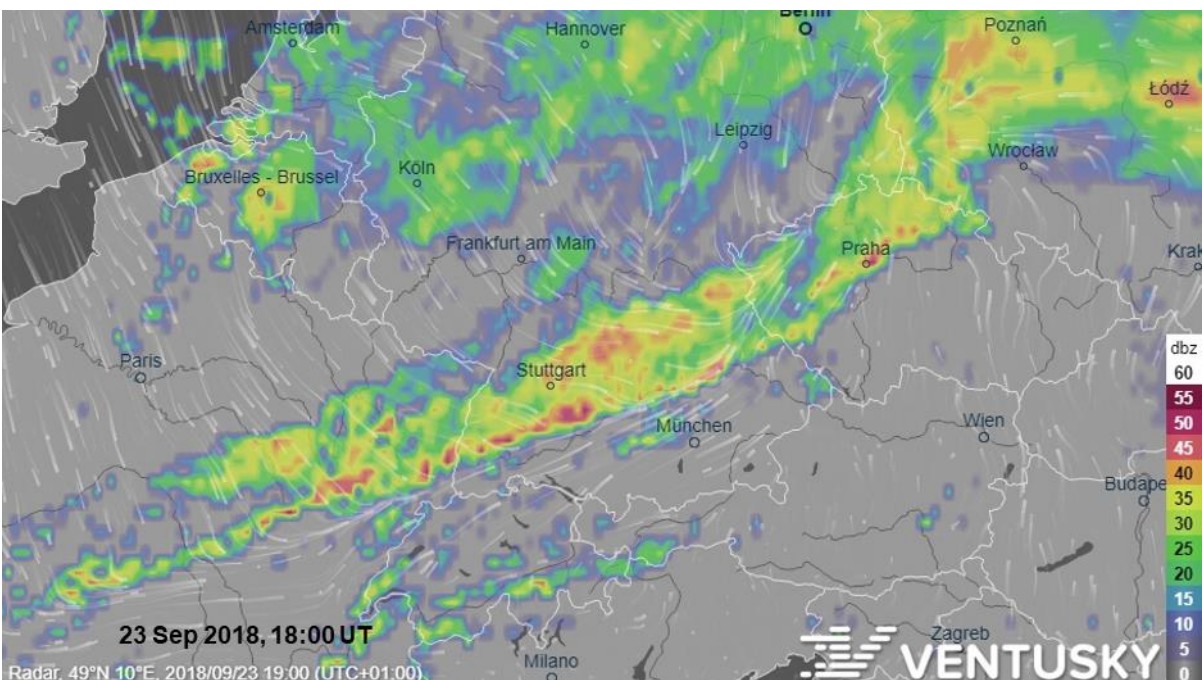

Fig. 4. Horizontal maximum projection of radar reflectivity from the European radar provided by VentuSky (https://www.ventusky.com). There is a rapid shift of the strong thunderstorms line (squall line) by 350 km per three hours. White lines illustrate the wind speed at ten meters above the ground. The colour scale indicates the radar reflectivity in 5 dBZ increments. The scale demonstrates exponential dependence of precipitation intensity [mm h$^{-1}$] on radar reflectivity [dBZ]. The value of 55 dBZ (dark purple) corresponds to the instantaneous value of intensity of convective precipitation 100 mm h$^{-1}$.

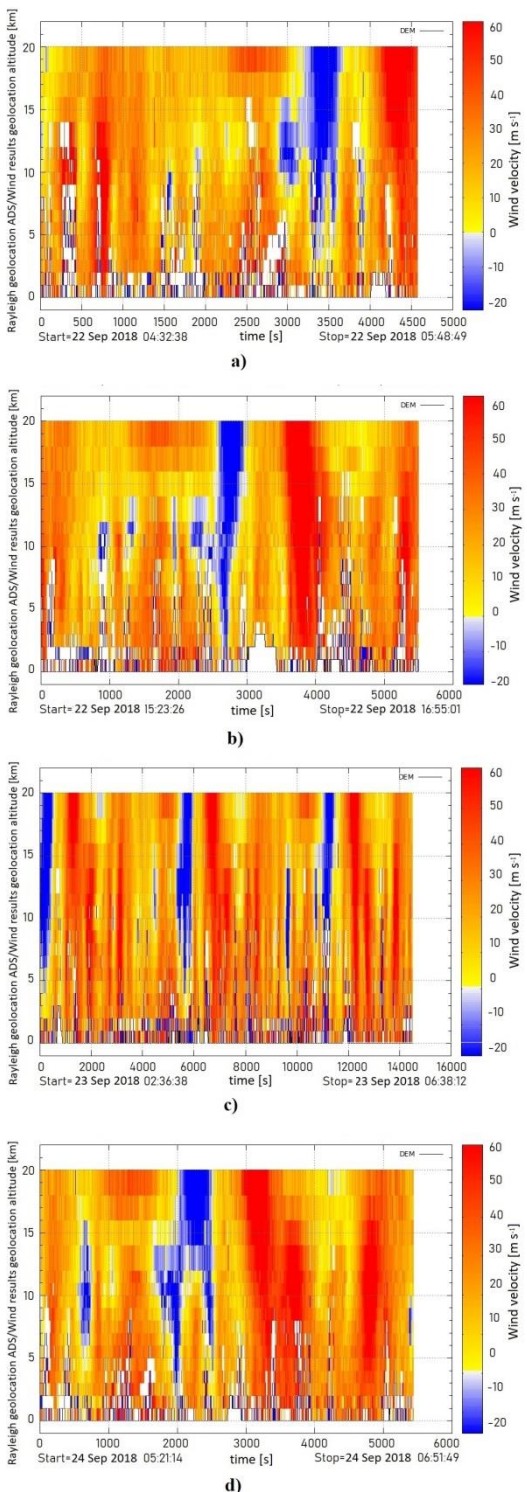

**Fig.5.** Aeolus Rayleigh scattering observations of wind profiles up to 20 km. The MDS
wind profiles (positive towards the instrument) measured during the orbit numbers 513 and 514
on 22–24 September 2018 between 04:32 and 06:51 UTC using the MDS Rayleigh scattering.
The blue areas indicate zones during which the wind speed variations introduced by the
Rayleigh response fluctuations are negative. Random Error (1 σ) is 1 m.sec$^{-1}$ at altitudes less
than 2 km, 2 m.sec$^{-1}$ between 2 and 16 km. Systematic Error (1 σ) is smaller than 0.7 m.sec$^{-1}$.

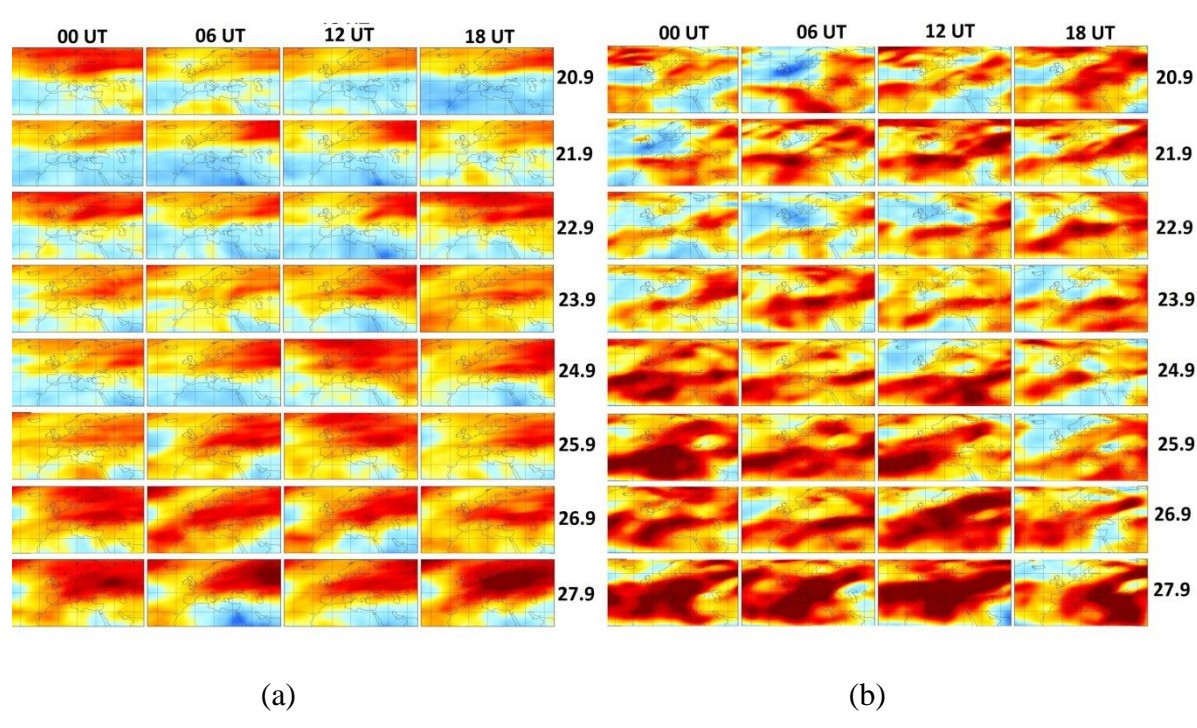

8                              (a)                                        (b)

**Fig. 6.** Stratospheric wind above Europe at 1hPa (panel a), and at 0.1hPa (panel b). Red color represents eastward wind, blue represents westward wind.

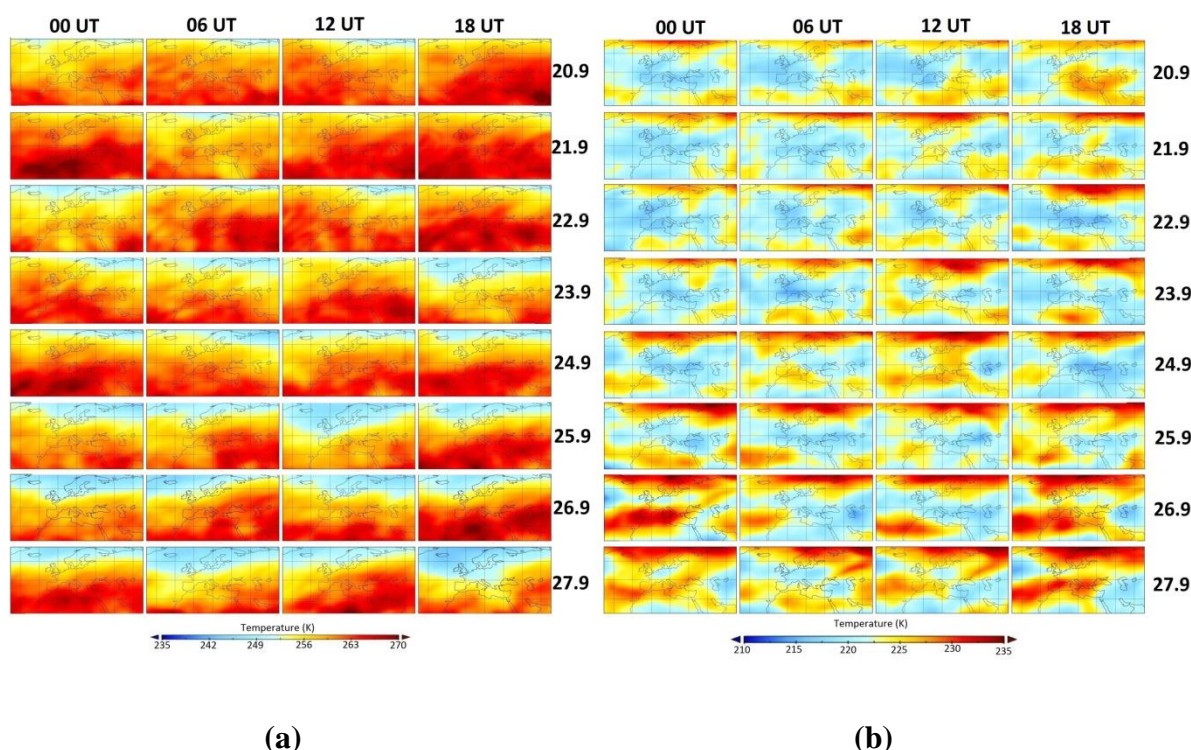

5                                      **(a)**                                          **(b)**

**Fig. 7.** Stratospheric temperature above Europe at 1hPa (panel a; temperature in the range 235 – 270 K) and at 0.1 hPa (panel b; temperature in the range 210 – 235 K).

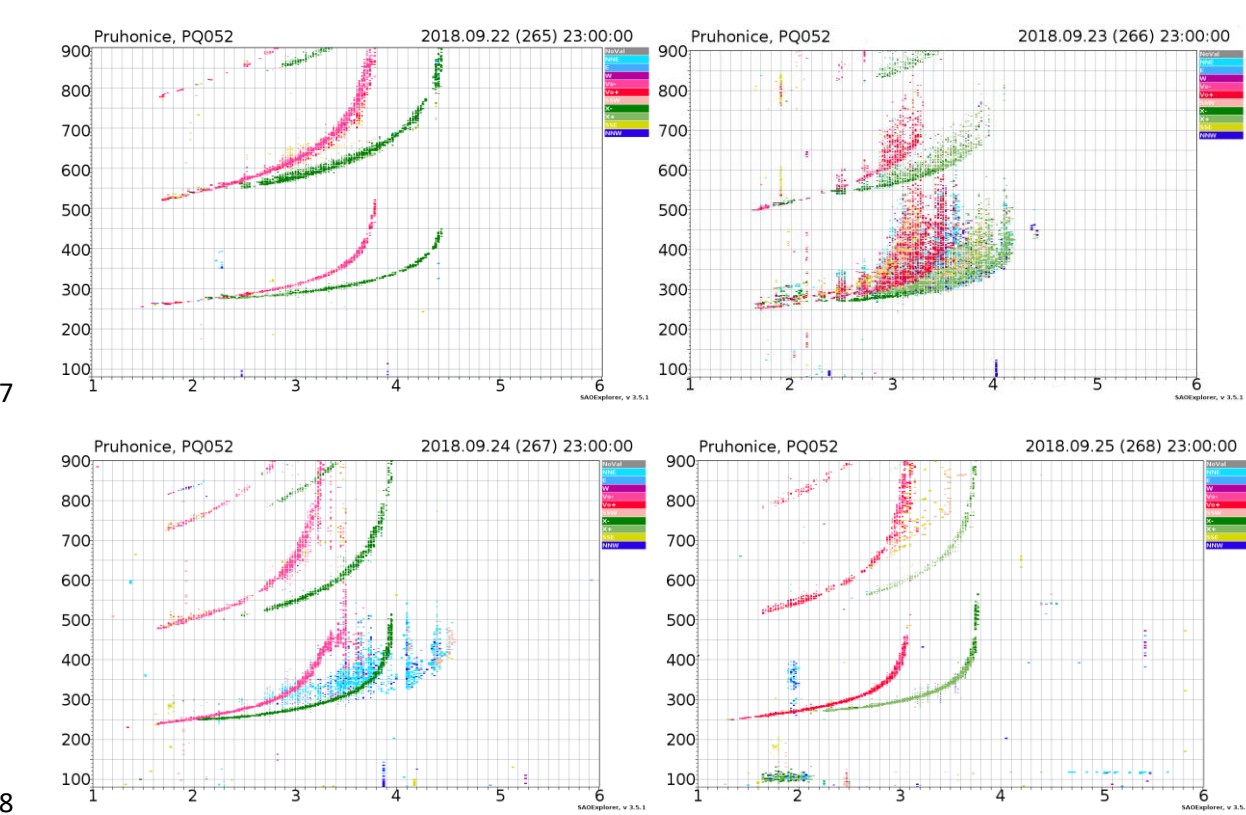

**Fig. 8.** Sequence of ionograms recorded during local night at around 23 UT presented in a
standard DPS 4D format. Particular color indicates direction of radio wave arrival. Vertical
echo of ordinary wave is marked by red color, while green color stands extraordinary wave.
Other colors represent non-vertical reflections. The strongest non-vertical echo comes from
North-North-East direction (light blue color).

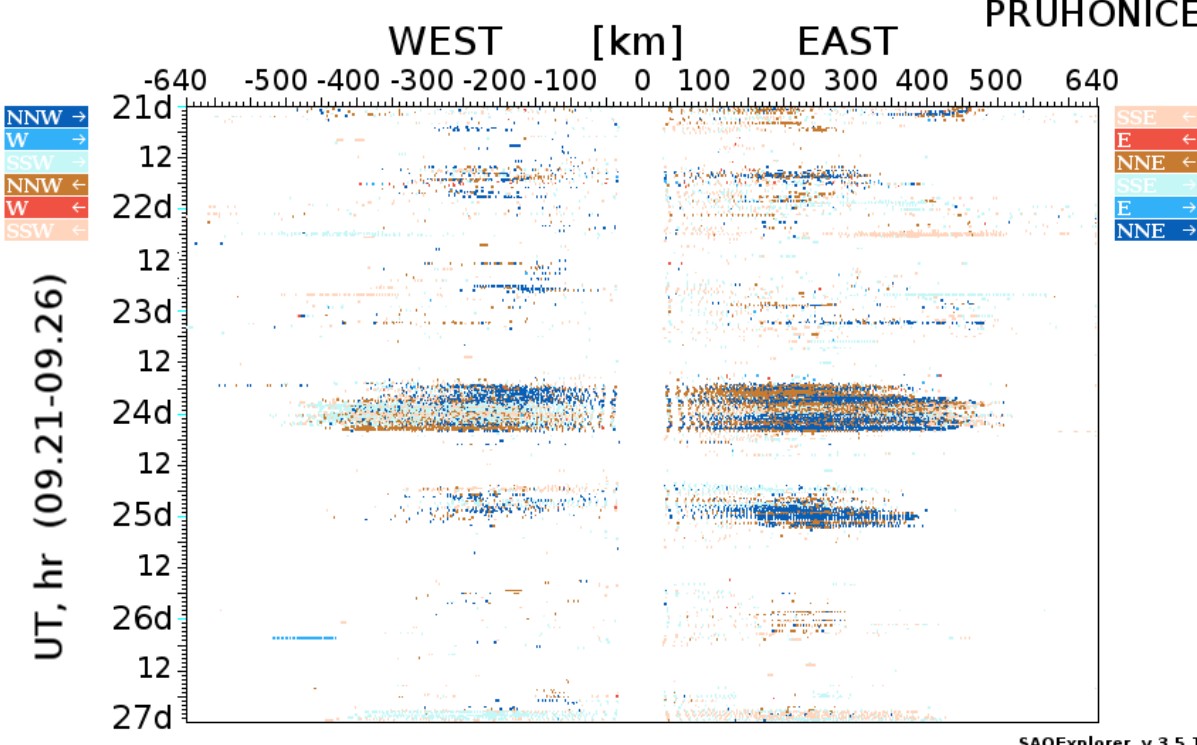

**Fig. 9.** Directogram – direction of plasma motion above the observation site - recorded on 21–
26 September presented in a standard DPS 4D format. Due to geometry of receiving antenna
field, DPS 4D identifies direction of the wave arrival. The directograms are constructed from
the raw ionograms. Each color correspond to particular antenna beam, hence the direction of
the arrival of oblique echo from large scale irregularities. It is denoted by particular color on
the left and right side of the diagram.

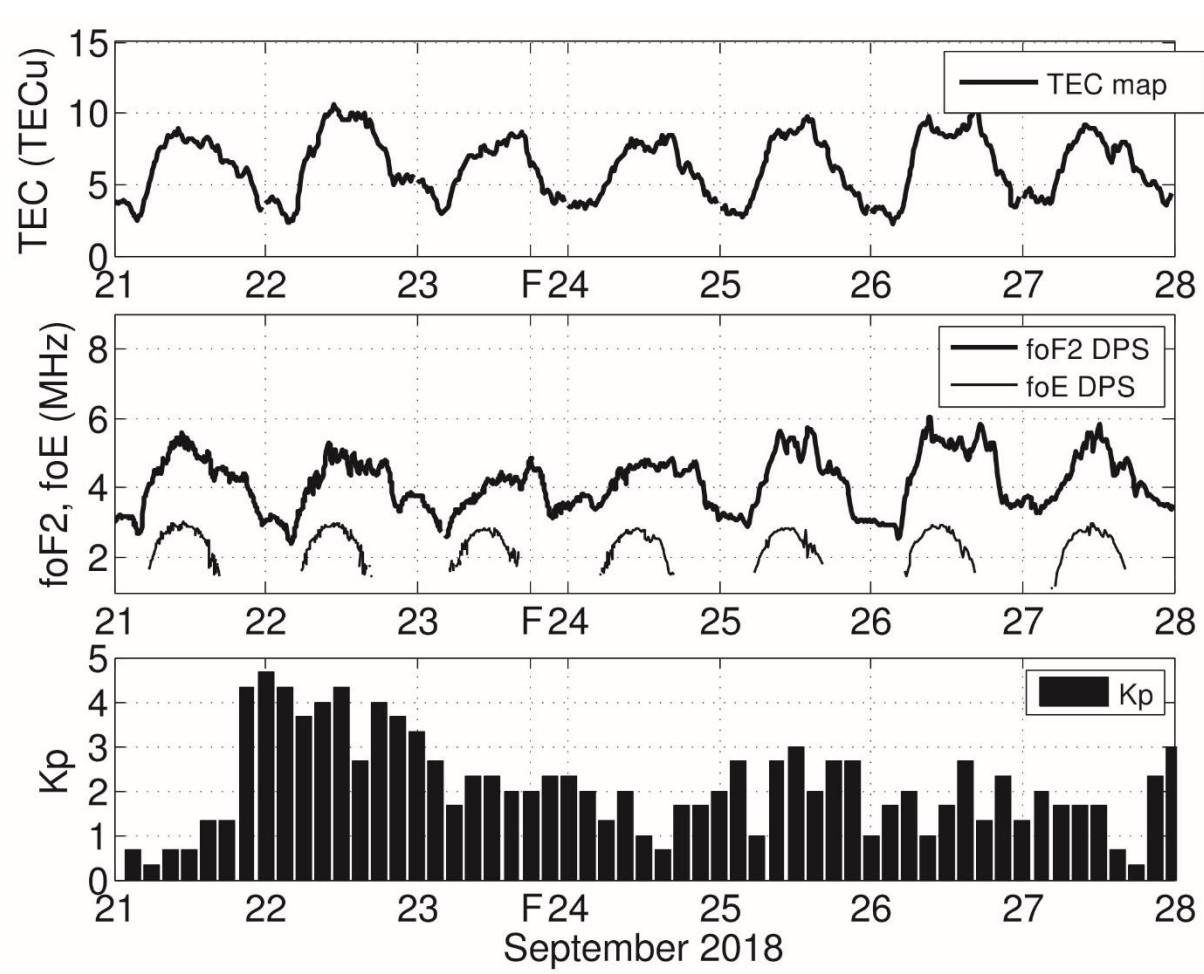

**Fig. 10.** Diurnal courses of Total Electron Content (upper panel), critical frequencies foF2 and foE (middle panel) above station Pruhonice ("F" denotes time of passage of the storm Fabienne over the station) and course of Kp index (bottom panel) during the observational period.

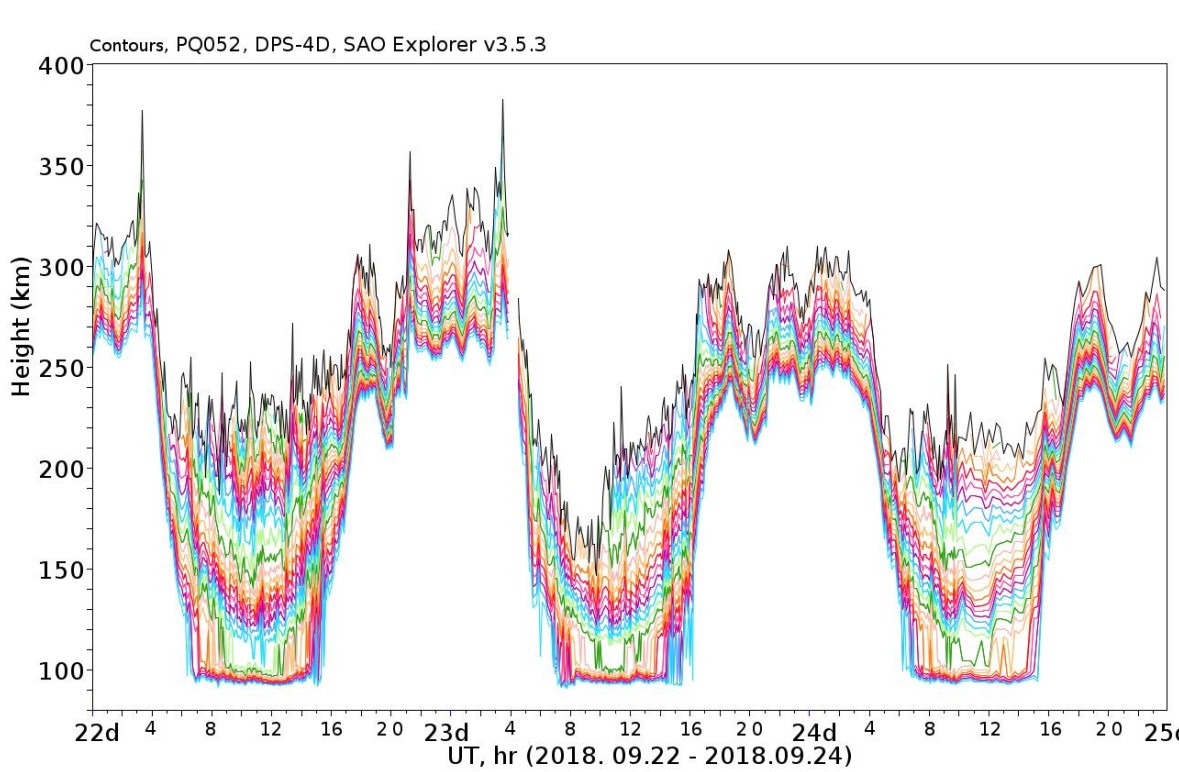

11 **Fig.11.** Variability of true-height reflection at fixed frequencies recorded during 22–24
12 September for frequency range 2–6 MHz with 0.1 MHz step, in a standard DPS 4D format.

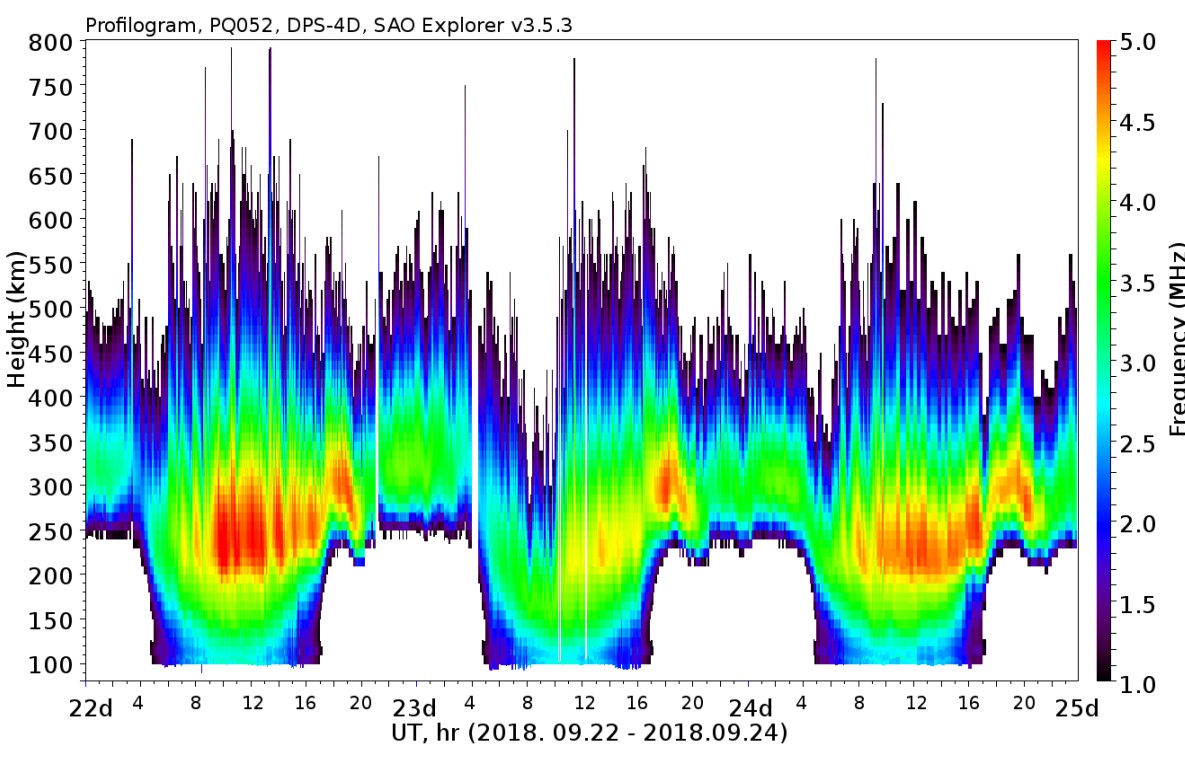

**Fig. 12.** Detail of profilograms in frequency – 22 September–24 September presented in a standard DPS 4D format.

**Fig.13.** Wavelet Power Spectra of critical frequency foF2 for 21–27 September, 2018. WPS is
normalized on each scale to corresponding scale maximum. Dark areas exceed 0.95 significance
level on each scale.

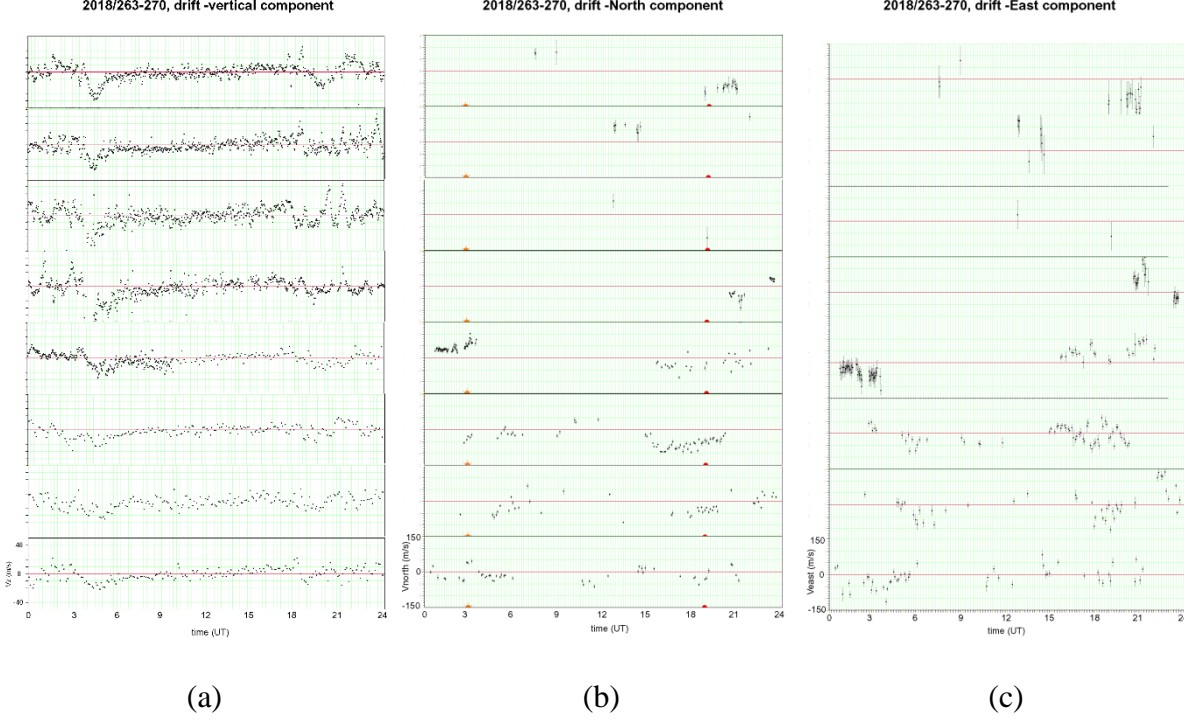

19            (a)                                (b)                                (c)

**Fig. 14.** Components of ionospheric plasma drift obtained by Digisonde DPS 4D during the studied event 20 September (top panel) – 27 September (bottom panel) – (a) vertical component; (b) North component; (c) East component presented in a standard DPS 4D format.

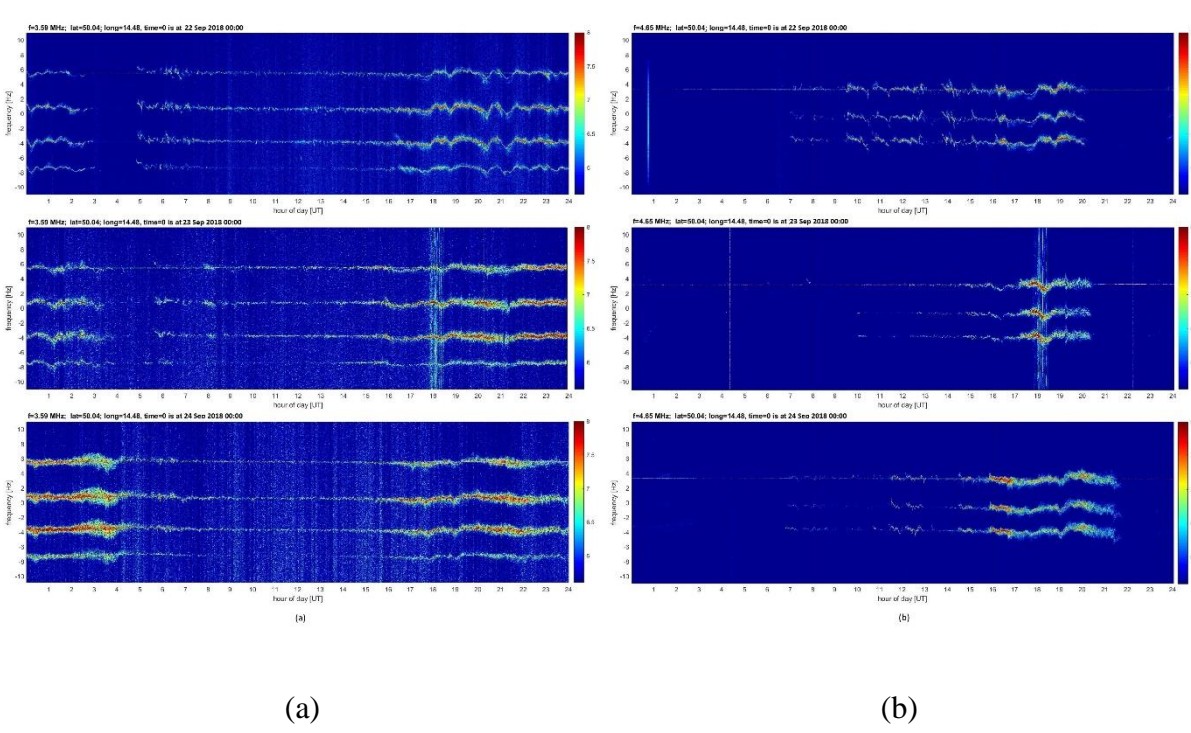

12                              (a)                                              (b)

**Fig. 15.** Continuous Doppler Sounding measurement on three consequent days 22–24
September on frequency 3.59 MHz (a) and 4.65 MHz (b).

