# Peer review of "Evidence of vertical coupling: Meteorological storm Fabienne"

_Annales Geophysicae, 2019_

## Referee Comment (RC1) · Anonymous Referee #1 · 22 Mar 2019

The presented paper undoubtedly corresponds to the profile of the journal AnnGeo and is aimed at studying the influence of powerful tropospheric processes on the dynamics of the overlying regions of the neutral atmosphere and the ionosphere. The subject of vertical coupling between the atmosphere layers is very relevant and has been actively discussed. The authors of the paper have written an informative review of a large number of scientific works on this subject, which certainly increases the level of scientific research. The influence of powerful tropospheric disturbances on the overlying atmosphere and the ionosphere has been repeatedly considered in the scientific literature

on the example of tropical cyclones, when the ionospheric reaction was estimated, for the most part, according to GPS-GLONASS monitoring. In the opinion of the reviewer, the absence of the analysis of these works in the introduction somewhat reduces the completeness of the approach used by the authors. Some of the methods used in the analysis of the ionospheric response to powerful tropical cyclones, and conclusions obtained from an analysis of the influence of tropical cyclones on the overlying layers, could be useful for this study. Perhaps the authors here are simply limited in size of manuscript. The main problem in reading the manuscript is the problem with the analysis of pictures, starting with Figure 6. If sufficiently detailed signatures are prepared for Figures 1–5, other illustrations are difficult to read. – Figures 6 and 7 according to the reanalysis of MERRA: it is not clear in what order they need to be analyzed (in rows or columns?), as the dates are difficult to distinguish even at high magnification, and in the captions of the figures and in the text itself there is no necessary explanation. In printed version this will be at all unreadable. – Figure 8: I recommend in the caption to make an explanation of which direction of signal arrival corresponds to the color of the track on the ionogram. These inscriptions are not readable in the figures. – Figure 9: I recommend to give explanations about what is displayed on the directograms, what physical quantity is represented. – Figure10: Perhaps here you should give the Kp or Ap index, otherwise the explanations of the level of geomagnetic activity in the text do not look convincing. – Figure13: There is no power scale. It is not clear where values are greater, where are less. – Figure14: The plots indicate the day of the year, it is necessary, at least, in the caption to indicate the date. – Figure 14. The plots show the numbers of days of the year. It is necessary, at least, in the figure caption to specify dates. – To improve perception, in all the figures it is desirable to indicate the moments of the front passage at the observation point.

Also there are a few comments to the references execution: 1. For the paper (Durand et al., 2004) (line 16 on page 5 and line 26 on page 9), the date of publication of the paper in the list of references is written 1989 (line 7-10 on page 17). 2. Similarly, for the paper (Roux et al., 2012) (line 4 on page 3), the date of publication of the paper in

the list of references is written 2002 (lines 36-38 on page 19). 3. In the text there are links to articles (Hook, 1971a) and (Hook, 1971b) (lines 18, 22 on page 3). However, in the list of references letters "a" and "b" does not put on. 4. For the paper "Sanders, F., 1986a..." (line 39 on page 19) it seems that the letter "a" should be removed and the year of publication should be moved to the end of the reference. 5. The article McDonald's et al. ... (line 46 on page 18) is not in alphabetical order in the reference list. 6. For the papers (Laštovička et al., 2012; Laštovička, 2012) (line 8 on page 3) there are no letters "a" and "b" both in the text of the paper and in the list of references (lines 5-8 on page 19).

After this minor revisions, I recommend the paper for publication.

Please also note the supplement to this comment:
https://www.ann-geophys-discuss.net/angeo-2019-26/angeo-2019-26-RC1-supplement.pdf

---

## Author Comment (AC1) · 18 Apr 2019

The authors would like to thank the reviewer for her/his constructive comments. 1. We agree with the reviewer comment that interesting results have been found based on the GPS-GLONASS monitoring. Very relevant papers have been published by Afraimovich et al. (2013), Ke et al. (2019), Yang and Liu (2016) and others dealing with tropical meteorological phenomena. Most of the papers address low latitude events, but we agree with the reviewer that these papers definitely have to be quoted in the review part of the paper. 2. In the revised version of the paper we provide better explanation

of the MERRA reanalyses results and improve results graphic representation (Figures 6 and 7) of the situation that capture time development of the situation before and after the storm passage over Europe. 3. In the revised version we provide suggested corrections on Figure 8. 4. On Figure 9 we show sequence of directograms as measured by Digisonde. The receiving antenna system of digisonde is able to identify the direction of the electromagnetic wave arrival. The directograms are constructed from the raw ionograms. Each color correspond to particular antenna beam, hence the direction of the arrival of oblique echo from large scale irregularities. The explaining text is added into the corrected version. 5. Due to length of the paper we have decided to show the plot of Kp index as it is available parameter from data center. According to reviewer suggestion, we add the plot into the revised version of the paper. We agree that it improves the paper. 6. Figure 13 corrected 7. Figure 14 corrected according to suggestions 8. Reference – corrected in the text and reference list. The reference Laštovička et al., 2012 and Laštovička, 2012, we have not corrected due to the fact that the first reference refers to paper written by team Laštovička Solomon and Qian, while the second one to the paper written by Laštovička only. We appreciate all the reviewr's comments and suggestion that help us to improve quality of the paper.

---

## Referee Comment (RC2) · Anonymous Referee #1 · 23 Apr 2019

After revision, I recommend the paper for publication.

---

## Referee Comment (RC3) · Anonymous Referee #2 · 9 May 2019

The authors reported a multi-instrument experiment to study the effects of tropospheric event on overlying neutral and ionized layers of the atmosphere. The topic is relevant since there are still many open questions in connection with the troposphere - upper atmosphere (stratosphere, mesosphere, termosphere, ionosphere) coupling mechanisms. The present study is especially interesting because the investigated meteorological event (storm Fabian) occurred during the recovery phase of a moderate geomagnetic storm. Therefore, it is a very good candidate to investigate the effects caused by both events on the ionosphere in the same time. It gives the opportunity to compare the importance of the troposphere - ionosphere coupling with the impact of the geomagnetic storm. The topic corresponds to the profile of the journal, especially to this special issue. However, I suggest to answer the following questions and comments before acceptance of the manuscript to publish.

General comments: 1. The authors give a very good review about the troposphere - upper atmosphere coupling in the introduction part. This very detailed summary/review could be complemented with a paragraph about the impact of the tropospheric events on the sporadic E layer because there are some very interesting papers which investigate and discuss this topic (e. g. Davis and Johnson 2005, Barta et al. 2017, Haldoupis 2018). 2. Page 8. line 17. "Around 15 UT the warm front brought light rain associated with stratiform clouds" It is not clear the date in this case for me. 3. Page 9. line 31-36: Please, discuss a little bit what we see on Fig. 5. and how it is related to the other observations. 4. Page 12. line 6-9: "Both values agree well through the studied interval and their matching can be explained by dominant contribution of F2 layer's electron contribution to the TEC and much less contribution of E layer's variability during studied days, even during the Fabienne event." Can you detail this explanation, please? Maybe it can be useful to show the variation of the foE parameter as well on Fig. 10. 5. Page 12. line 16-19.: "Geomagnetic disturbance started on 21 September at 21 UT. Frequency foF2 during night falls much faster than it is typical. Then foF2 oscillates and remain below 3.5 MHZ till almost noon when rapidly increases." The second part is related to the variation on the night 22/23 September? Please, indicate the date, because it is not clear for me. 6. Page. 13. line 40-41: "The spectral content changed with time and was different during the strong storm event compare to preceding and following day. " We can't see similar effect at 4.65 MHz. Can you give an explanation for that? 7. Page 14. line 30-32. "According to the evolution of Kp index and ionospheric plasma parameters (TEC and foF2) ionosphere was already in the recovery phase of the geomagnetic storm. Nevertheless, the observed disturbances are induced both by geomagnetic storm and convective activity in the lower laying atmosphere. " Can you discuss in more details the convective activity effect on the

TEC and/or foF2 and how it appears on Fig. 10.? I can not distinguish its impact from the geomagnetic storm in the case of these two parameters. General comments to the Figures: Unfortunately, it is very difficult to see the following figures (especially in print version): Fig. 3, Fig 6. and 7., Fig. 8, Fig. 14. Please, indicate the letters a, b, c etc. on the subplots where it is necessary. In some cases, when you show sequence of pictures it could help if you indicated the dates (in row e.g. on Fig 6 and 7) and the time (above the columns e.g. on Fig. 6 and 7) Minor comments: 1. Sometimes the line spacing change in the manuscript: e.g. page 3. line ...7-8 / 9-10....; page 7. line ... 30-31 / 32-33 2. Page 3. line 16. The effects of gravity waves on in the ionosphere: in should be deleted 3. Page 4. line 1. On the longer term-term scale: one term should be deleted 4. Page 6. line 25. Kouba et al. (2008)

Please also note the supplement to this comment:
https://www.ann-geophys-discuss.net/angeo-2019-26/angeo-2019-26-RC3-supplement.pdf

---

## Author Comment (AC2) · 27 May 2019

The authors thank to reviewer for her/his constructive comments that helps to improve quality of the paper. We agree with the reviewer that implementation of all her/his suggestion help the readability of the paper.

General comments: 1. As suggested by the reviewer, we have implemented brief information about influence of meteorological systems on formation and behavior of sporadic E and included suggested references. 2. Corrected. The time of the front

passage is explained in detail. 3. Meteorological interpretation is added into the text. 4. Larger discussion is added into the text according suggestion. Course of critical frequency foE is added into figure 10. 5. Corrected. Sentence added to make clear when the variability is described. 6. Discussion of observation on frequencies frequency 3.59 MHz and 4.65 MHz is added. 7. We try to provide insight into the situation. For that the discussion is enlarged. However, splitting these two effects is not possible since all our observation provides resulting variability.

Comments to figures: Fig. 3, Fig.6. and Fig.7, Fig.8, Fig. 14 – corrected according to suggestion.

Fig.6 and Fig.7. – corrected and reorganized. Minor comments: corrected according to suggestions.

---

## Author Comment (AC4) · 6 Jul 2019

On behalf of all authors, I would like to thank the reviewer for her/his valuable comments and suggestions that helped us to improve the paper significantly. Corrections made in the text are provided further together with corrected figures.

Sincerely, Petra Koucka Knizova

Here are all the requested corrections that we made in the text: 1. We have added text related to GLONASS/GPS:

[Figure]

GPS satellite measurements are promising tools for monitoring ionospheric changes connected with severe weather systems. Recently the analyses of scintillation S4 index in relation to four tropical cyclones (Yasi in 2011, Marcia in 2015, Debbie in 2017 and Marcus in 2018) were presented by Ke et al. (2019). They found intensification of scintillation effects mostly above the tropical cyclone path and attributed them to the electric field perturbation and consequent plasma bubble generation. Within COSMIC GPS data, Yang and Liu (2016) has found significant peak in radio occultation scintillation events during the passage of tropical cyclone Tembin (2012) during quiet geomagnetic or solar aktivity and attributed the observed effect to the gravity waves generated in the lower atmosphere by the cyclone. Afraimovich et al. (2013) published large review of GPS/GLONASS studies of the ionospheric response to natural and antropogenic processes and phenomena. Paper focuses on wide range of ionospheric forcing and corresponding ionospheric variability detected in principle within Total Electron Content (TEC) and F2 layer critical frequency foF2. In relation to tropical cyclones (Katrina, Rita and Wilma) occurring in 2005 they reported increase of wave-like activity in gravity-wave period range mainly in the range 20 to 60 minutes and intensification of TEC variations along the satellite path close cyclone. The zones of disturbances were found to form during hurricane stage of the cyclone (Afraimovich et al., 2013).

2. Explanation of MERRA results:

Figure 6a shows zonal wind at 1 hPa for Europe region from 20 September at 00 UT to 27 September at 18 UT. On the sequence there is well seen weak eastward wind in middle Europe and westward wind in south Europe which is typical situation for this period. Shortly before storm Fabienne (23 Sep 00 and 6 UT) easterly wind became stronger and replace westerly wind in the south (because of incoming waves from troposphere) and remain easterly for the following several days in whole studied area. At 0.1 hPa we can see changes from westerly to easterly wind shortly after Fabienne (24.9. 00 and 06 UT). We do not register any significant changes before because wave from the troposphere need some time to reach 0.1 hPa. Strong easterly wind remains in whole Europe again for several days after storm. The stratosphere needs some time for changing/restoration dynamics to normal situation because of wave disturbances which remains in inversion condition (temperature increase with altitude) much longer than in other layers. That is why we can observe strong eastward wind not only during the storm but for several days after storm in whole Europe as well.

4. Explanation text of the ionospheric measurement

An example of multi-beam ionograms measured by DPS-4D is shown in the Figure 8. There is a sequence of ionograms recorded during four consequent days around 23 UT. The receiving antenna system of the digisonde is able to identify the direction of the electromagnetic wave arrival. The information about the reflected wave arrival is included in the raw ionograms. Further, from sequence of raw ionograms the general plasma motion is constructed and presented as the directogram. Each color correspond to particular antenna beam, hence the direction of the arrival of oblique echo from large scale irregularities.

We have corrected all the figures as suggested.

Please also note the supplement to this comment:
https://www.ann-geophys-discuss.net/angeo-2019-26/angeo-2019-26-AC4-supplement.pdf

**Supplement:**

[Figure]

(a)                                                  (b)

**Fig. 6.** Stratospheric wind above Europe at 1hPa (panel a), and at 0.1hPa (panel b). Red color represent eastward wind, blue represents westward wind.

[Figure]

**(a)** **(b)**

**Fig. 7.** Stratospheric temperature above Europe at 1hPa (panel a; temperature in the range 235

– 270 K) and at 0.1 hPa (panel b; temperature in the range 210 – 235 K).

[Figure]

**Fig. 8.** Sequence of ionograms recorded during local night at around 23 UT presented in a
standard DPS 4D format.

[Figure]

[Figure]

**Fig. 9. D**irectogram recorded on 21–26 September presented in a standard DPS 4D format.

[Figure]

**Fig. 10.** Diurnal courses of Total Electron Content (upper panel), critical frequencies foF2 and foE (middle panel) above station Pruhonice ("F" denotes time of passage of the storm Fabienne over the station) and course of Kp index (bottom panel) during the observational period.

[Figure]

**Fig.11.** Variability of true-height reflection at fixed frequencies recorded during 22–24
September for frequency range 2–6 MHz with 0.1 MHz step, in a standard DPS 4D format.

[Figure]

**Fig. 12.** Detail of profilograms in frequency – 22 September–24 September presented in a
standard DPS 4D format.

[Figure]

[Figure]

(a)                     (b)                     (c)
**Fig. 14.** Components of ionospheric plasma drift obtained by Digisonde DPS 4D – (a) vertical
component; (b) North component; (c) East component presented in a standard DPS 4D format.

[Figure]

         (a)                                 (b)

**Fig. 15.** Continuous Doppler Sounding measurement on three consequent days 22–24 September on frequency 3.59 MHz (a) and 4.65 MHz (b).

---

## Author Response (AR1)

The authors would like to thank the editor and both anonymous reviewers for their valuable suggestions and comments that helped us to significantly improve the manuscript. All the changes are marked by yellow color in the text.

Sincerely,

Petra Koucka Knizova (on behalf of all the authors)

Answer to Reviewer 1 comments:

1. In the opinion of the reviewer,the absence of the analysis of these works in the introduction somewhat reduces thecompleteness of the approach used by the authors. Some of the methods used in the analysis of the ionospheric response to powerful tropical cyclones, and conclusions obtained from an analysis of the influence of tropical cyclones on the overlying layers,could be useful for this study. Perhaps the authors here are simply limited in size ofmanuscript.

*We agree with the reviewer comment that interesting results have been found based on the GPS-GLONASS monitoring. Very relevant papers have been published by Afraimovichet al. (2013), Ke et al. (2019), Yang and Liu (2016) and others dealing with tropical meteorological phenomena. Most of the papers address low latitude events, but we agree with the reviewer that these papers definitely have to be quoted in the reviewpart of the paper. – Paragraf added on pg. 4*

GPS satellite measurements are promising tools for monitoring ionospheric changes connected with severe weather systems. Recently the analyses of scintillation $S_4$ index in relation to four tropical cyclones (Yasi in 2011, Marcia in 2015, Debbie in 2017 and Marcus in 2018) were presented by Ke et al. (2019). They found intensification of scintillation effects mostly above the tropical cyclone path and attributed them to the electric field perturbation and consequent plasma bubble generation. Within COSMIC GPS data, Yang and Liu (2016) has found significant peak in radio occultation scintillation events during the passage of tropical cyclone Tembin (2012) during quiet geomagnetic or solar aktivity and attributed the observed effect to the gravity waves generated in the lower atmosphere by the cyclone. Afraimovich et al. (2013) published large review of GPS/GLONASS studies of the ionospheric response to natural and antropogenic processes and phenomena. Paper focuses on wide range of ionospheric forcing and corresponding ionospheric variability detected in principle within Total Electron Content (TEC) and F2 layer critical frequency foF2. In relation to tropical cyclones (Katrina, Rita and Wilma) occurring in 2005 they reported increase of wave-like activity in gravity-wave period range mainly in the range 20 to 60 minutes and intensification of TEC variations along the satellite path close cyclone. The zones of disturbances were found to form during hurricane stage of the cyclone (Afraimovich et al., 2013).

2. Illustrations are difficult to read.

Figures 6 and 7 according to the reanalysis of MERRA: it is not clear in what order they need to be analyzed (in rows or columns?), as the dates are difficult to distinguish even at high magnification, and in the captions of the figures and in the text itself there is no necessary explanation. In printed version this will be at all unreadable

*In the revised version of the paper we provide better explanation of the MERRA reanalyses results and improve results graphic representation (Figures 6 and 7) of the situation that capture time development of the situation before and after the storm passage over Europe.*

Figure 6a shows zonal wind at 1 hPa for Europe region from 20 September at 00 UT to 27 September at 18 UT. On the sequence there is well seen weak eastward wind in middle Europe and westward wind in south Europe which is typical situation for this period. Shortly before storm Fabienne (23 Sep 00 and 6 UT) easterly wind became stronger and replace westerly wind in the south (because of incoming waves from troposphere) and remain easterly for the following several days in whole studied area. At 0.1 hPa we can see changes from westerly to easterly wind shortly after Fabienne (24.9. 00 and 06 UT). We do not register any significant changes before because wave from the troposphere need some time to reach 0.1 hPa. Strong easterly wind remains in whole Europe again for several days after storm. The stratosphere needs some time for changing/restoration dynamics to normal situation because of wave disturbances which remains in inversion condition (temperature increase with altitude) much longer than in other layers. That is why we can observe strong eastward wind not only during the storm but for several days after storm in whole Europe as well.

Figure 8: I recommend in the caption to make an explanation of which direction of signal arrival corresponds to the color of the track on the ionogram. These inscriptions are not readable in the figures.
*Caption extended*

Particular color indicates direction of radio wave arrival. Vertical echo of ordinary wave is marked by red color, while green color stands extraordinary wave. Other colors represent non-vertical reflections. The strongest non-vertical echo comes from North-North-East direction (light blue color).

Figure 9: I recommend to give explanations about what is displayed on the directograms, what physical quantity is represented.
*Caption extended*

Directogram – direction of plasma motion above the observation site - recorded on 21–26 September presented in a standard DPS 4D format. Due to geometry of receiving antenna field, DPS 4D identifies direction of the wave arrival. The directograms are constructed from the raw ionograms. Each color correspond to particular antenna beam, hence the direction of the arrival of oblique echo from large scale irregularities. It is denoted by particular color on the left and right side of the diagram.

Figure10: Perhaps here you should give the Kp or Ap index, otherwise the explanations of the level of geomagnetic activity in the text do not look convincing.
*Kp added into Figure 10*

Figure13: There is no power scale. It is not clear where values are greater, where are less.
*Caption extended*

Wavelet Power Spectra of critical frequency foF2 for 21–27 September, 2018. WPS is normalized on each scale to corresponding scale maximum. Dark WPS areas exceed 0.95 significance level on each scale.

Figure14: The plots indicate the day of the year, it is necessary, at least, in the caption to indicate the date. – Figure 14. The plots show thenumbers of days of the year. It is necessary, at least, in the figure caption to specifydates.

*Caption extended*

Components of ionospheric plasma drift obtained by Digisonde DPS 4D during the studied event 20 September (top panel) – 27 September (bottom panel) – (a) vertical component; (b) North component; (c) East component presented in a standard DPS 4D format.

To improve perception, in all the figures it is desirable to indicate the momentsof the front passage at the observation point.

*We have improved quality and caption of all the figures.*

Also there are a few comments to the references execution:

1. For the paper (Durandet al., 2004) (line 16 on page 5 and line 26 on page 9), the date of publication of the paper in the list of references is written 1989 (line 7-10 on page 17).

*Corrected*

2. Similarly, forthe paper (Roux et al., 2012) (line 4 on page 3), the date of publication of the paper in the list of references is written 2002 (lines 36-38 on page 19).

*Corrected*

3. In the text there arelinks to articles (Hook, 1971a) and (Hook, 1971b) (lines 18, 22 on page 3). However,in the list of references letters "a" and "b" does not put on.

*Corrected*

4. For the paper "Sanders,F., 1986a..." (line 39 on page 19) it seems that the letter "a" should be removed andthe year of publication should be moved to the end of the reference.

*Corrected*

5. The article McDonald's et al....(line 46 on page 18) is not in alphabetical order in the reference list.

6. For the papers (Laštovička et al., 2012; Laštovička, 2012) (line 8 on page 3)there are no letters "a" and "b" both in the text of the paper and in the list of references(lines 5-8 on page 19).

*The papers differ in author teams, hence there is no letters "a" and "b" necessary in the references.*

Answer to Reviewer 2 comments:

General comments:

1. The authors give a very good review about the troposphere -upper atmosphere coupling in the introduction part. This very detailed summary/review could be complemented with aparagraph about the impact of the tropospheric events on the sporadic E layer because there are some very interesting papers which investigate and discuss this topic (e. g. Davis and Johnson 2005, Barta et al. 2017, Haldoupis 2018).

*Text added:*

Tropospheric convective systems are often connected with strong lightning. Possibility of thunderstorm influence on ionosphere has been already suggested by Bhar and Syam (1937). In general, two principal mechanisms are proposed. First mechanism presumes gravity waves generated by thunderstorm to propagate up to ionospheric heights. Second mechanism involves generation of electrical discharges in the E region above the storm. Applying superposed epoch analyses, Davis and Johnson (2005) reported statistically significant intensification and decent in altitude of midlatitude sporadic E layer directly above thunderstorm. Different observational result showing decrease of critical frequency of sporadic E has been reported by Barta et al. (2017). Mechanism involved in the coupling between thunderstorm lightning and ionosphere is very complicated and not well understood yet. The limitations of generally accepted mechanisms are discussed in detail in the paper Haldoupis (2018).

2. Page 8. line 17. "Around 15 UT the warm front brought light rain associated with stratiform clouds"It is not clear the date in this case for me.
*Text corrected:*

Around 15 UT on September 23, the warm front brought light rain associated with stratiform clouds.

3. Page 9. line 31-36: Please, discuss a little bit what we see on Fig. 5. and how it is related to the other observations.
*Text added:*

The accuracy is limited by the design of the instrument. In all the comparisons we consider this aspect. Single Doppler Wind Lidar is able to measure both Mie scattering from particles and aerosols, and Rayleigh scattering from the upper atmosphere molecules. This study uses the Rayleigh scattering measurement with random error ($1\sigma$) is 1 m.sec-1 at altitudes less than 2 km, 2 m.sec$^{-1}$ between 2 and 16 km. Systematic error ($1\sigma$) is in this case smaller than 0.7 m.sec$^{-1}$ (Durand et al., 2004).

In the two upper panels of Figure 5a and 5b we may see situation before the storm on 22 September. At heights above 10 km there is area where the satellite registers opposite direction of the wind compare to surrounding regions (marked by blue color). Figure 5c represents situation of early morning of 23 September before the cyclone Fabienne has entered the area of measurement site. Calming of the windflow caused by temperature daily cycle is clearly visible. Figure 5d shows the post storm effect on 24 September. The area of opposite wind direction detected by satellite Rayleight scattering is lifted up to heighs of 15 km. The measurements at the time of Fabienne storm passage above the measurement site is not available due to satellite trajectory, however from the satellite records before and after the cyclone passage indicate extremely high speed changes within troposphere and lower stratosphere.

4. Page 12. line 6-9: "Both values agree well through the studied interval and their matching can be explained by dominant contribution of F2 layer's electron contribution to the TEC and much less contribution of Elayer's variability during studied days, even during the Fabienne

event." Can you detail this explanation, please? Maybe it can be useful to show the variation of the foE parameter as well on Fig. 10.

*Text added:*

While TEC and foF2 show significant decrease in reaction to minor geomagnetic disturbance, there is no clear change in course and shape of critical frequency of E layer foE Figure 10 (middle panel) except of very short wave-like variability on 23 September before Fabienne storm passage above the observational site. On 23 September, maximum of foE reaches same values as on preceding and following days. Most of the variability is observable within time series of TEC and foF2 and both parameters agree well through the studied interval. Their matching can be explained by dominant contribution of F2 layer's electron content contribution to the TEC and much less contribution of E layer's variability during studied days, even during the Fabienne event.

*Figure 10 corrected*

5. Page 12. line 16-19.: "Geomagnetic disturbance started on 21 September at 21 UT. Frequency foF2 during night falls much faster than it is typical. Then foF2 oscillates and remain below 3.5 MHZ till almost noon when rapidly increases." The second part is related to the variation on the night 22/23 September? Please, indicate the date, because it is not clear for me.

*Text corrected:*

Geomagnetic disturbance started on 21 September at 21 UT. Frequency foF2 during night falls much faster than it is typical. Then during night 22-23 September, critical frequency foF2 falls even faster, oscillates and remains below 3.5 MHz till almost noon when rapidly increases.

6. Page. 13. line 40-41: "The spectral content changed withtime and was different during the strong storm event compare to preceding and following day. " We can't see similar effect at 4.65 MHz. Can you give an explanation for that?

*Text enlarged:*

The spectral content changed with time and was different during the strong storm event compared to preceding and following day. During afternoon hours on 22 September, CDS registers clear sharp echo with wave-like fluctuations. On 23 September on both frequencies we have observed sudden increase of noise at 18 UT that could indicate arrival of acoustic wave packet from the frontal border. After that, stronger and blurred echo compared to 22 September is registered on both frequencies. Wave-like fluctuations are not detected within the signal on 3.59 MHz and 4.65 MHz. On both frequencies (better pronounced on 4.65 MHz on Figure 15 b), there are apparent coincidental drops in frequency at 18 UT. Blurred strong echo was observed until around 4 UT on 24 September. In the afternoon hours on 24 September, recorded CDS echo remains slightly blurred but it is significantly weaker. The occurrence of stronger echo on CDS sounding on 3.59 MHz in the interval 18 UT (23 September) till 4 UT (22 September) corresponds to the increased wave activity on directograms and detection of plasma flow on both North and East plasma drift components. The trace of 4.65 MHz is limited due to diurnal course of foF2. Hence the changes of in the CDS signal can be discussed only till 20 UT. Signal detected on 23 September is significantly stronger with respect to preceding and following days, especially in the part that corresponds to the frequency drop at 18 UT.

7. Page 14. line 30-32. "According to the evolution of Kp index and ionospheric plasma parameters (TEC and foF2) ionosphere was already in the recovery phase of the geomagnetic storm. Nevertheless, the observed disturbances are induced both by geomagnetic storm and convective activity in the lower laying atmosphere." Can you discuss in more details the convective activity effect on the TEC and/or foF2 and how it appears on Fig. 10.? I can not distinguish its impact from the geomagnetic storm in the case of these two parameters.

*Text added:*

*In the introduction part:*

Model study (Pedatella, 2018) demonstrated variability of the response of the atmosphere and ionosphere system to one particular storm when the internal variability characterized by the ensemble standard deviation is introduced. The study shows that implementation of arbitrary internal atmospheric variability leads to the geomagnetic storm occurring under a different, though climatically similar, atmospheric state for each ensemble member. The study has found that variability leads to uncertainty typically 20%–40% with locallized regions exceeding 100%. It clearly shows that large-scale features of the storm are reproduced well and while small-scale characteristics of the response are dependent on lower atmosphere variability. Hence neglecting of the lower atmosphere may lead to significant complication in the geomagnetic storm interpretation.

*In the Discussion part:*

The observed variability of the parameters TEC, foF2 and foE on Figure 10 is caused jointly by the minor geomagnetic disturbance and atmospheric waves associated with Fabienne storm. It is practically impossible to distinguish what part of the variability belongs to the particular forcing. Ionospheric vertical sounding has, unfortunately, limitations and provides integral information about resulting behavior of the atmosphere. However, in addition to the time of flight of the electromagnetic wave the DPS 4D equipment recorded additional parameters of the reflected wave from ionosphere. Variability of critical frequency foF2 must be interpreted together with complete ionogram record. As it is demonstrated in the Figure 8, there is well seen change of the ionogram pattern through experiment. Ionograms recorded on 22 September (type on panel a) are usually recorded when the reflection plane is practically flat while ionograms recorded on 23 September (type on panel b) are measured when reflection planes are significantly undulated. Such situations occur in association with atmospheric wave activity. Hence, this additional information can be used to slightly untangle effects of the geomagnetic disturbance and convective activity. Taking into account the course of foF2 and TEC together with change of ionogram reflection patterns, we suppose, that dominant effect of the geomagnetic disturbance is pronounced as decrease of foF2 and TEC, while short term wave-like variability around mean course associated with spread echo occurrence on ionograms can be attributed to the convective activity in the lower atmosphere.

*In the conclusion part:*

Regarding results of model study (Pedatella, 2018) we attribute general decrease in foF2 and TEC to the geomagnetic forcing (longer-term, negative storm scenario) and significant increase in wave-like activity (short-term, wave-like activity) to the convective system forcing.

General comments to the Figures:

Unfortunately, it is very difficult to see the following figures (especially in print version):Fig. 3, Fig 6. and 7., Fig. 8, Fig. 14. Please, indicate the letters a, b, c etc. on the subplots where it is necessary. In some cases, when you show sequence of pictures it could help if you indicated the dates (in row e.g. on Fig 6 and 7) and the time (above the columns e.g. on Fig. 6 and 7)

*The authors hope that the Figures are corrected sufficiently*

Minor comments:

1. Sometimes the line spacing change in the manuscript: e.g. page 3. line ...7-8 / 9-10....; page 7. line ... 30-31 / 32-33

*Corrected*

2. Page 3. line 16.The effects of gravity waves on in the ionosphere: in should be deleted

*Corrected*

3. Page 4. line 1. On the longer term-termscale: one term should be deleted

*Corrected*

4. Page 6. line 25. Kouba et al. (2008)

*Corrected*

Editor´s suggestion:

*We have added text in the Introduction part:*

[revised manuscript text omitted]